# Parallel loss of sexual reproduction in field populations of a brown alga sheds light on the mechanisms underlying the emergence of asexuality

Masakazu Hoshino [1,6], Guillaume Cossard[1], Fabian B. Haas [1], Emma I. Kane[1], Kazuhiro Kogame[2], Takahiro Jomori[3], Toshiyuki Wakimoto [3], Sylvain Glemin[4,5] & Susana M. Coelho [1] ✉

Sexual reproduction is widespread, but asexual lineages have repeatedly arisen from sexual ancestors across a wide range of eukaryotic taxa. The molecular changes underpinning the switch to asexuality remain elusive, particularly in organisms with haploid sexual systems. Here we explore independent events of loss of sex in the brown alga *Scytosiphon*, examine the proximate and evolutionary mechanisms involved, and test the importance of sexual conflict on gene expression changes following loss of sex. We find that asexual females ('Amazons') lose ability to produce sex pheromone and, consequently, are incapable of attracting males, whereas they gain rapid parthenogenic development from large, unfertilized eggs. These phenotypic changes are accompanied by convergent changes in gene expression. Decay of female functions, rather than relaxation of sexual antagonism, may be a dominant force at play during the emergence of asexuality in haploid sexual systems. Moreover, we show that haploid purifying selection plays a key role in limiting the accumulation of deleterious alleles in Amazons, and we identify an autosomal locus associated with the Amazon phenotype. The sex chromosome, together with this autosomal locus, may underlie the switch to obligate asexuality in the Amazon populations.

The shift from sexual reproduction to asexual reproduction by parthenogenesis has occurred in several eukaryotic lineages[1,2] but how the loss of sex affects the phenotype of the asexuals, the molecular changes involved in the switch and the genomic consequences of asexuality are largely unknown. Asexual lineages often occupy terminal nodes in phylogenies and are thought to be evolutionary dead ends[3]. According to theory, they are expected to have lower genetic diversity, reduced levels of adaptation and generally lower fitness than their sexual relatives[4,5]. However, empirical reports indicate that asexual populations frequently have similar diversity and rates of adaptation as their sexual relatives (for example, see refs. 6,7). It is still unclear, therefore, whether transitions to asexuality consistently bring the genomic changes that are predicted.

The extent to which the switch to asexuality is associated with large changes in gene expression is still debated. Asexual lineages may have a selective advantage over sexual lineages, as they are relieved from 'intralocus sexual conflict'[7,8]. Sexual conflict or sexual antagonism occur when the two sexes have conflicting strategies to optimize their fitness for reproduction. If there is prevalent conflict between the two sexes concerning gene expression (for example, if the expression

of a gene gives the male a fitness advantage but the female a fitness disadvantage), a shift to full asexuality would be expected to be accompanied by increases in expression of genes with primarily female functions (because they would be relieved from sexual antagonism) and decreases in expression of genes with primarily male functions. Yet, studies in animals found a masculinization of gene expression in asexuals[9,10]. Thus, how the transition to asexuality impacts sexual conflict for optimal gene expression remains unclear. Moreover, the detailed molecular mechanisms underlying the shift to asexuality are largely unknown. Although hundreds of genes appear to be differentially expressed in sexuals versus asexuals[9,11,12], very few loci that are genetically associated with asexual states have been found[13,14].

Shifts to asexuality from sexual ancestors take place also in brown algae (brown seaweeds). These largely marine, multicellular eukaryotes belong to the stramenopile group, being very distantly related to animals and plants[15]. Their biology is largely underexplored; however, in recent years they have become key model organisms to study the evolution of sex determination, reproductive systems and complex life cycles[16]. Most brown algae alternate between multicellular male and female haploid gametophytes (the so-called gametophyte generation) and a sporophyte (diploid) generation ('haplo-diplontic' life cycle)[16–19] (Fig. 1a). Importantly, gametophyte and sporophyte generations are independent and free living, with the haploid stage being morphologically complex. These features offer unique opportunities to study the effect of haploid selection on the evolution and maintenance of asexuality. Sex is determined at meiosis (not at fertilization, as in XY and ZW systems) depending on whether haploid spores inherit a female (U) or a male (V) chromosome[20,21]. Spores grow into multicellular female or male gametophytes, which at maturity produce female or male gametes (Fig. 1a). Most brown algae are broadcast spawners and gamete attraction and fertilization occur via pheromone production by female gametes[22]. The majority of these pheromones are unsaturated, non-functionalized acyclic and/or alicyclic C11 hydrocarbons, and the biochemical pathway that underlies their production is complex and still elusive[23]. Broadcast spawning organisms, where sexual selection is constrained to act after gamete release, offer highly tractable models for evaluating gamete-level mate choice and sexual conflict[24].

In parallel with this sexual life cycle, brown algae may have an asexual life cycle in which unfused female (and occasionally male) gametes undergo parthenogenesis to develop into adult multicellular individuals[25–30]. The events underlying haploid spore production from (presumably haploid) parthenosporophytes is currently elusive but are thought to involve apomeiosis (either a non-reductive meiosis or endoreduplication followed by meiosis)[26]. Mechanistically, therefore, the basis of the transition from sex to asexuality in haplo-diplontic organisms is clearly distinct compared with animals and plants; for example, the transition happens during the haploid stage, and meiosis does not need to be modified because parthenogenesis occurs in the haploids. Given their dramatic differences in terms of life cycle, the mechanisms and consequences of loss of sex (and transition to fully asexual reproduction) in haplo-diplontic organisms are expected to be fundamentally different from those described for animals and, as comparative models, they provide an opportunity to explore the universality (or uniqueness) of this key biological transition. Previous work on the experimental model brown alga *Ectocarpus* found that parthenogenesis is controlled by the sex locus and involves two additional autosomal loci[28], highlighting the key role of the sex chromosome as a major regulator of asexual reproduction in this organism. However, the exact molecular mechanism underpinning the transitions to obligate parthenogenesis remains to be determined.

Transitions to obligate asexuality have also been observed in field populations of the brown alga *Scytosiphon*[31]. Such asexual populations (referred to as 'Amazon' populations) and their closely related, sexual ancestral populations provide a unique opportunity to study the transition from sexual reproduction to obligate asexuality in haplo-diplontic

species. Moreover, the fact that transitions to asexual reproduction occur independently and repeatedly in these organisms allows us to investigate the conservation of the mechanisms underlying the switch to asexuality.

Here we report the identification as well as phenotypic and genomic characterization of asexual populations of a brown alga with haploid sex determination, and we contrast these populations with their sexual ancestors. We sequenced the gamete transcriptomes of several individuals from closely related pairs of asexual and ancestral sexual populations of two *Scytosiphon* species. This approach allowed us to test whether the pattern of gene expression consistently changes with the evolution of asexuality and whether there is a 'feminization' of gene expression in asexual females, consistent with a release from sexual antagonism. We found that a female-specific trait (pheromone production) decays in Amazon populations, and there was a concomitant optimization of their capacity for asexual reproduction, by gamete parthenogenesis. These phenotypic modifications were accompanied by defeminization and masculinization of gene expression in *S. promiscuus*. We identified a significant number of genes exhibiting convergent expression changes, exceeding that expected by chance, including genes within the female sex-chromosome regions, which supports a role for sex-specific loci in the emergence of asexuality. Importantly, we show that haploid purifying selection may limit the accumulation of deleterious alleles in Amazons. A genome-wide comparative analysis of female and Amazon populations revealed one locus, encoding an EF-hand-domain-containing protein that was fully associated with asexuality. This locus may play a role in signalling pathways related to the Amazon phenotype. Overall, by identifying changes occurring in multiple independent transitions to asexuality, our findings provide insights into the molecular basis of asexual reproduction in organisms with haploid sex determination and suggest that the evolutionary path to asexuality may be constrained, probably requiring repeated changes in the same key genetic pathways.

## Results

### Amazon populations of *Scytosiphon* diverged repeatedly from sexual ancestors

We had previously identified natural populations of *S. lomentaria* composed exclusively of females[31,32]. These Amazon populations appear to grow exclusively by asexual reproduction, in which female gametophytes develop following apomeiosis of parthenosporophytes formed upon parthenogenesis of unfertilized female gametes (Fig. 1a). To investigate the prevalence of Amazon populations in the *Scytosiphon* genus, we sampled populations of the species *S. promiscuus* at four sites (Koinoura, Koi; Shioyazaki, Sz; Inoshiri, Ii and Im) on the coast of Japan (Supplementary Tables 1 and 2, and Fig. 1b) and determined their sex ratios by using sex markers[20,33] and crossing experiments with reference strains. In total, 156 gametophytes were identified as *S. promiscuus* (Supplementary Table 2). Two of the populations were sexual (Im and Koi), that is, both males and females were present and two populations had a sex ratio of 0:1 (male:female), so were composed exclusively of females (Sz and Ii) (Supplementary Table 2). These observations thus indicate that Amazon populations of *Scytosiphon* are relatively common and widespread, as are those of *S. lomentaria*[31].

To investigate whether the Amazon populations arose from ancestral sexual lineages across the distribution range, we examined the phylogenetic relationships between the *Scytosiphon* populations. We chose a subset of individuals from each population and sequenced RNA from gametes for each lineage (Supplementary Table 1). RNA-seq data were used to build a phylogenetic tree, which showed a clear separation of the *S. lomentaria* and *S. promiscuus* lineages, and revealed that Amazons emerged from their sexual ancestors repeatedly and independently in each species, with probably two independent origins of Amazon lineages within *S. promiscuus* (Fig. 1c), even if we cannot fully exclude that genes involved in asexuality could have been introduced

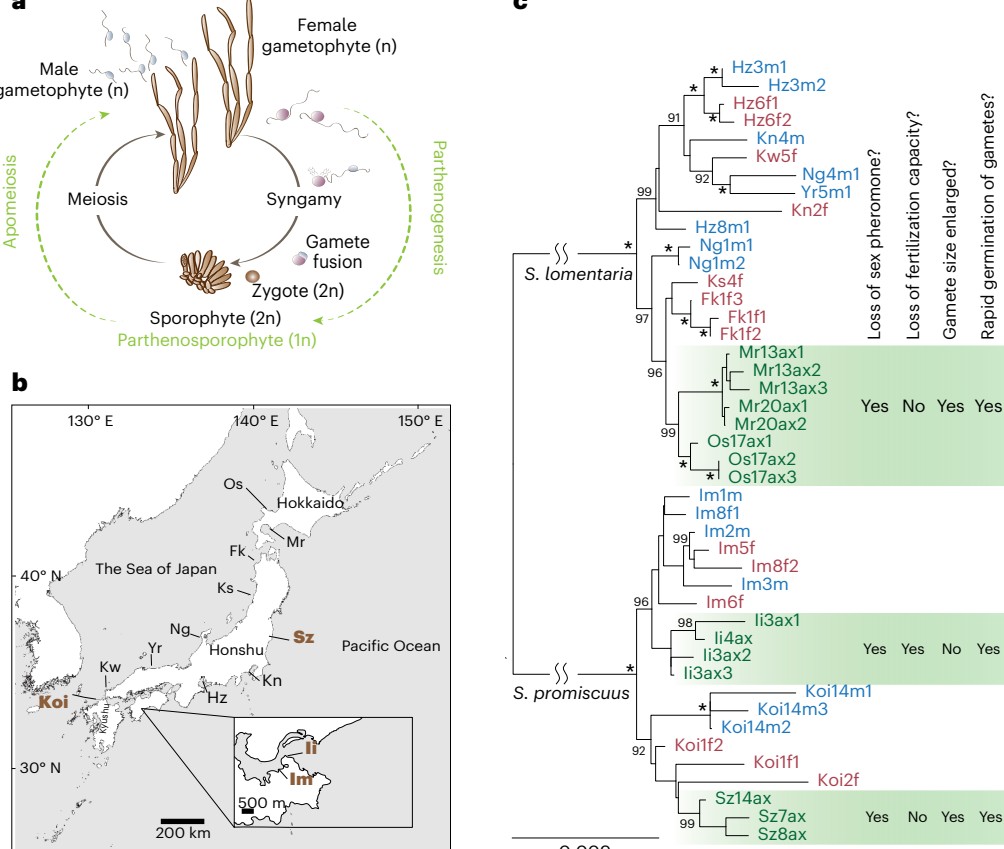

**Fig. 1 | Amazon populations of *Scytosiphon* diverged from sexual ancestors ~1.5 Ma. a**, Haploid-diploid life cycle of *Scytosiphon* sp. *Scytosiphon* sp. has a heteromorphic life cycle, where generations of multicellular and macroscopic dioicous isomorphic gametophytes alternate with generations of microscopic multicellular discoid sporophytes. *Scytosiphon* sp. male and female gametes are almost indistinguishable in terms of size, with female gametes being slightly bigger than male gametes, but they show clear sexual dimorphism in behaviour and physiology: female gametes settle on the substratum sooner than male gametes and secrete a sex pheromone that attract male gametes[110]. Note that unfused female gametes may enter a parthenogenetic cycle (in green) in the absence of fertilization and develop into parthenosporophytes which, after apomeiosis, reconstitute the gametophyte generation. Amazon populations use exclusively the parthenogenetic/asexual part of the life cycle. Adapted from ref. 19 **b**, Geographical location of the different populations. Detailed location

is given in Supplementary Table 1. The four sampled populations of this study are highlighted in brown bold font. **c**, Phylogenetic relationships between the studied populations. The maximum likelihood tree was based on concatenated DNA sequences of 53 single-copy orthologues. Numbers on branches indicate bootstrap values from maximum likelihood analysis. The asterisks indicate branches with full support (100). Only bootstrap values >90 are shown. Female and male samples from sexual populations are represented in red and blue colour, respectively, and Amazon female samples are in green colour. Summary of gamete behaviour phenotypes scored in Amazons (green) compared to sexual females. Pheromone was measured either by olfaction in a blind test or by GC–MS (Methods). Attachment of male flagellum was determined by microscope observation (blind test). Gamete fusion was scored 5 min after mixing male and female gametes by observation of zygote production (confirmed by the presence of two eyespots), as in ref. 111.

---

from one lineage to the other via introgression. On the basis of two independent approaches and datasets, namely, phylogenetic dating using mitochondrial and chloroplast markers and demographic modelling using nuclear data (Methods), we estimated the divergence of asexual populations from sexual ancestors ~1.5 million years ago (Ma) in both *S. lomentaria* and *S. promiscuus* (Extended Data Fig. 1 and Supplementary Table 3).

### Decay of pheromone production and optimization of asexual traits in Amazon populations

Similar to most brown algae, *Scytosiphon* species are broadcast spawners: when the gametophytes reach maturity, female and male gametes are simultaneously released into the surrounding seawater. Female gametes rapidly settle and produce a pheromone, a $C_{11}H_{16}$ compound[34] that attracts male gametes. This compound can be detected by gas chromatography/mass spectrometry (GC–MS) or by an olfaction test[34]. Male gametes swim towards female gametes using their two flagella, scan the membrane surface of the female gametes with their anterior

flagella, then the gametes fuse to form a zygote, which develops into a (diploid) sporophyte (Fig. 2a).

Gametes from *S. lomentaria* Amazon populations do not produce pheromone[32]. We tested whether gametes from *S. promiscuus* Amazon populations were likewise unable to produce the sexual pheromone. GC–MS analysis detected a $C_{11}H_{16}$ compound in female gametes from sexual populations of *S. promiscuus* but failed to detect this pheromone or other $C_{11}$ hydrocarbons in samples from any of the Amazon strains of this species.

To test whether gametes from *S. promiscuus* Amazon populations can still fuse with male gametes from sexual populations in laboratory conditions, we recorded the behaviour of Amazon gametes when the male gametes were present at high density to 'force' the encounter between them. In these conditions, most Amazon Ii populations did not form zygotes (Fig. 2c), indicating little or no fertilization. In contrast, related sexual populations (Im) produced a large proportion of zygotes. We also noticed that male gamete flagella did attach to the membranes of the Amazon Ii gametes, but this recognition resulted in

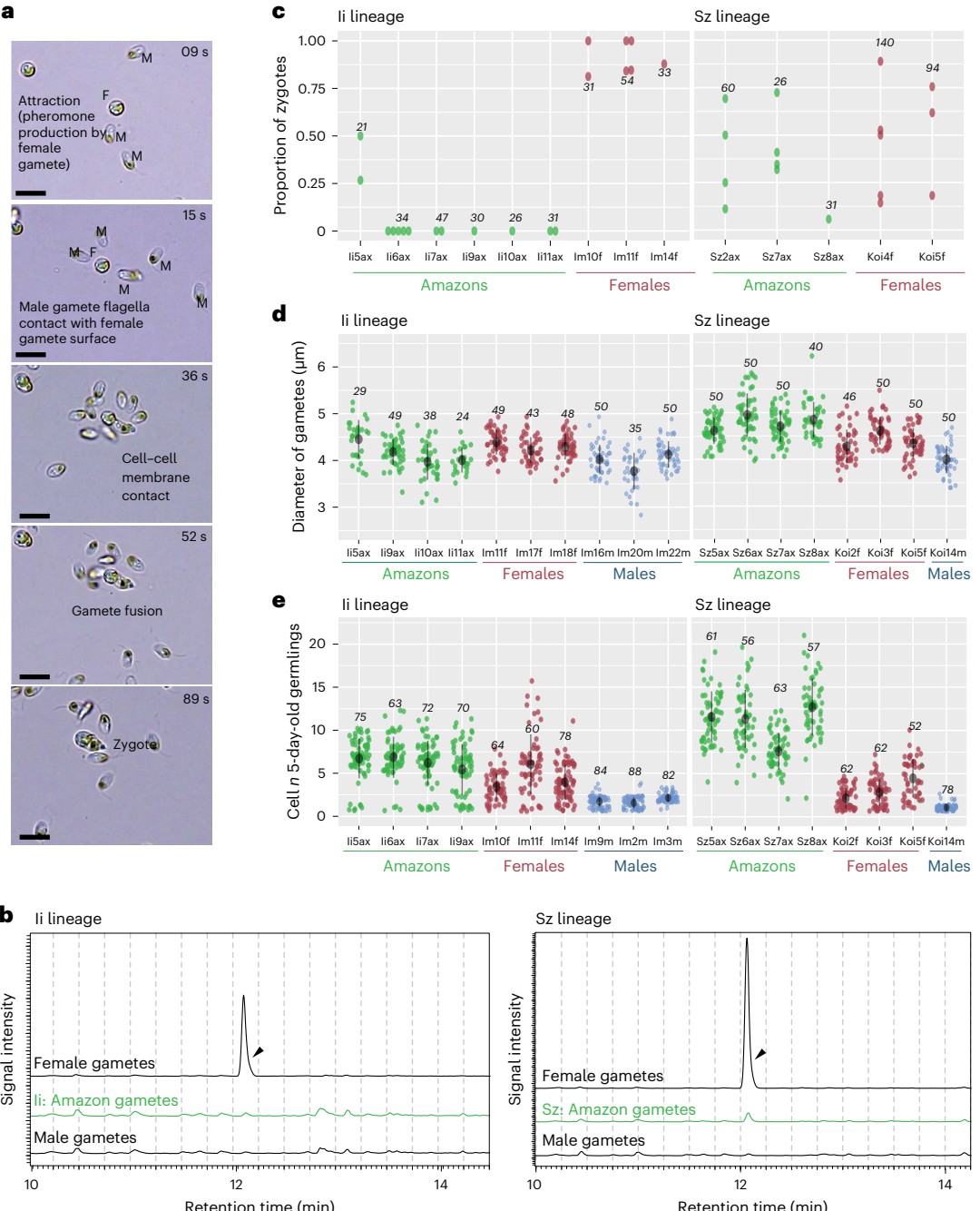

**Fig. 2 | Decay of pheromone production and fertilization capacity, and optimization of asexual traits in two Amazon lineages of *S. promiscuus* (Sz and Ii). a**, Time lapse of fertilization in a sexual population of *S. promiscuus*. Time (in seconds) is indicated in the top right of each photo. M, male gamete; F, female gamete. The first step is the attraction of male gametes towards female gametes via the production of the pheromone by the latter. Once male gametes are at the vicinity of the female gamete, they scan the surface of the female membrane using their flagella. The last step in the process is cell membrane contact and gamete fusion. **b**, Representative GC–MS analyses of the volatile compounds released from gametes: extracted ion chromatograms (MIC) = 91–92 *m/z*. The compound predicted as hormosirene (6-[(Z)-1-butenyl]-1,4-cycloheptadiene) was detected with high intensity at ~12.1 min from female gametes of sexual populations (arrowheads) but not from female gametes of Amazon populations. **c**, Fertilization success of female gametes from sexual versus Amazon populations. We scored the proportion of zygotes arising in a controlled cross (see Methods for details). Each point on the graph represents the fertilization success of one independent cross. The number of scored gametes is indicated by the italicized number on each plot. **d**, Diameters of gametes in each line of *S. promiscuus*. The number of scored gametes is indicated by the italicized number on each plot and each dot represents one gamete. The mean ± s.d. is shown as a point range (black) in each plot. **e**, Parthenogenetic capacity of gametes, illustrated by the number of cells in developing 5-day-old germlings. Female gametes may develop parthenogenetically into haploid parthenosporophytes if they do not encounter a male gamete. Non-fused gametes from Amazon lines develop faster than unfused gametes from sexual females. See also Supplementary Tables 5 and 6 for detailed statistical analysis.

gamete fusion in only one sample, Ii5ax (Fig. 2c). By contrast, Amazon Sz gametes formed as many zygotes as did the related sexual (Koi) female gametes in the conditions of high male gamete density (Fig. 2c,

and Supplementary Tables 4 and 5, Generalized Linear Mixed Model (GLMM)), indicating effective fertilization. The ability of Amazon gametes to recognize and fuse with the male gamete membrane was

unaffected in all populations except in population Ii in which, despite the high density of male gametes, no gamete fusion occurred (Fig. 2c). This suggests that the genetically female Amazon Ii population produces 'zoids' that cannot fuse with gametes of the opposite sex; it is fully asexual. The zoids from the other Amazon populations are affected only in their capacity to produce pheromone.

We performed a morphometric analysis of the gametes of Amazon and sexual populations to look for differences between them. *S. promiscuus* Amazon gametes tended to be bigger than female gametes from sexual populations, significantly so for population Sz (Fig. 2d and Supplementary Table 5, GLMM model). Moreover, the parthenogenetic capacity of the Amazon gametes, determined by the germination rate of unfused gametes after 24 h, was significantly greater than that of their closest sexual populations (GLMM model, Supplementary Table 5). Amazon parthenogenetic germlings were morphologically more developed than their sexual counterparts after 5 days in culture, as indicated by the number of cells they contained (Fig. 2e and Supplementary Table 5, GLMM model).

Our observations suggest that genetically female Amazon populations of both *S. promiscuus* and *S. lomentaria* do not produce a sex pheromone to attract male gametes. Despite the convergent decay of this sexual trait, Amazon gametes of both species retain their ability to recognize male gametes and fuse with them, except for Amazon gametes from the Ii population. This indicates that they have not become fully, obligatory, asexual, although in natural conditions they most probably never have sex because there are no males. Concomitant with loss of pheromone production, Amazon gametes undergo parthenogenesis more rapidly and grow faster than female sexual gametes, therefore, they appear to be optimized for asexual reproduction.

**Transcriptome landscape in males, females and Amazons**

To determine whether the phenotypic changes in the Amazon gametes were accompanied by modifications in their transcriptomic landscape when compared with male and female gametes, we used RNA-seq to measure transcript abundance in male and female gametes from the sexual populations and in Amazon gametes (Supplementary Tables 6 and 7). To map the transcriptomes, we generated high-quality whole-genome assemblies for both *S. lomentaria* and *S. promiscuus* by using a combination of long- and short-read sequencing (Supplementary Tables 6 and 7).

The number of genes expressed in the gametes was 11,378 in *S. lomentaria* (57% of the genome) and 12,809 in *S. promiscuus* (61% of the genome; Supplementary Tables 5 and 6). In principal component analysis (PCA) using all samples from female sexual and Amazon populations, samples clustered by species and by population; the females and Amazons did not form distinct clusters (Fig. 3a). It appears, therefore, that there is no extensive modification of transcription when all genes are compared between sexual females and Amazon. However, when PCA was performed separately for each species, and male, female and Amazon samples were included, the Amazon populations clustered together (Fig. 3b). This implies that the transcriptomes of Amazon populations are more closely related to each other than to their phylogenetically closest female populations.

Our phenotypic analyses, described above, revealed that Amazon gametes have lost a key female-specific trait, the ability to produce pheromone. To examine whether sex-biased gene expression underlies this phenotypic 'defeminization', we analysed the RNA-seq data with DEseq2 to identify gene sets that are expressed differentially in males versus females in the sexual strains (that is, sex-biased genes (SBG)). The analysis considered only genes that displayed at least a 2-fold change (FC) in relative expression between the sexes (Supplementary Table 7; FC > 2, $P_{adj}$ < 0.05). Less than 1% of the expressed genes in *S. lomentaria* and *S. promiscuus* gametes were sex-biased genes (Fig. 3c and Supplementary Table 6), consistent with the overall low level of sexual dimorphism in brown algae[35]. We found similar

proportions of male-biased genes and female-biased genes in both species (Supplementary Tables 7–9, and Fig. 3c). Functional enrichment analysis of sex-biased genes highlighted functions related to catabolism (peptidase activity) specifically in female-biased genes in both species (Extended Data Fig. 2 and Supplementary Table 10). In other species with UV sex chromosomes, sex-biased genes evolve faster than unbiased genes[36]. We tested whether this was the case for *Scytosiphon* sp. by comparing the evolutionary rates (dN/dS) of 4,747 expressed single-copy orthologues (SCO) in *S. lomentaria* and *S. promiscuus* that we identified by using OrthoFinder (Supplementary Table 10). For each species, we compared the evolutionary rates of SCO that have sex-biased expression with those of SCO whose expression is unbiased. The dN/dS values of female-biased genes were significantly greater than those of the unbiased genes, indicating a faster evolutionary rate, whereas we saw no evidence for faster evolution of male-biased genes (Extended Data Fig. 3).

To investigate how the expression of these sex-biased genes was modified in Amazon populations, we analysed the RNA-seq data for all the sex-biased genes in each species by hierarchical clustering in a heat map. Overall, the sex-biased gene transcriptomes of females and Amazon samples in both species were more similar to each other than they were to male samples; however, the Amazon samples clustered together and formed a separate cluster from female samples, particularly in *S. promiscuus* (Fig. 3d). This suggests that sex-biased gene expression in Amazons differs in some ways from that of a fully functional female.

The median level of female-biased gene expression (measured as $\log_2$(TPM + 1)) in Amazon females was significantly lower than that of sexual females (Wilcoxon test, *P* = 0.049 and *P* = 0.010 in *S. lomentaria* and *S. promiscuus*, respectively; Fig. 3e), suggesting that these genes are transcriptionally 'defeminized' in Amazons. By contrast, male-biased genes were all expressed at a higher level, or a similar level, in Amazons when compared with sexual females (Wilcoxon test, *P* = 0.003 and *P* = 0.02 in *S. lomentaria* and *S. promiscuus*, respectively; Fig. 3e and Extended Data Fig. 4), suggesting that Amazon gametes are transcriptionally 'masculinized'. Levels of expression of SBG were comparable between sexual lineages within each species, so that the observed pattern of defeminization and masculinizaton is specific to the transition to asexuality in Amazons of both species (Extended Data Figs. 4 and 5). Further analysis (see Methods for detail) confirmed a bona fide defeminization and masculinization of gene expression in *S. promiscuus* Amazons, although regression to the mean effects in *S. lomentaria* Amazons could not be excluded. The transcriptome defeminization/masculinization observed in *S. promiscuus* Amazons is consistent with the strong phenotypic changes, specifically in the Ii population which has lost the ability to fuse with male gametes.

Together, these findings indicate that transition to asexual reproduction in Amazons involves systematic changes in sex-biased gene expression in gametes, leading to defeminization and masculinization of gene expression, when compared with females.

To further characterize the changes in expression of sex-biased genes upon transition to asexuality, we calculated the mean expression of female-biased, male-biased and unbiased genes in females and Amazons of *S. lomentaria* and *S. promiscuus* from all the population samples and correlated expression of genes in the asexual samples with expression of the orthologous genes in the female sample by calculating the Pearson correlation coefficients. A correlation coefficient of 1 indicates a very similar level of expression, whereas correlation coefficients <1 indicate increasing differences in expression compared with ancestral sexuals. We found that the levels of unbiased gene expression in asexual samples were very similar to those in the sexual samples, whereas female-biased and male-biased gene expression in asexual samples were very different from those in the sexual samples (Fig. 3f). The effect was more marked for the female-biased genes than for the male-biased genes in most populations. In other

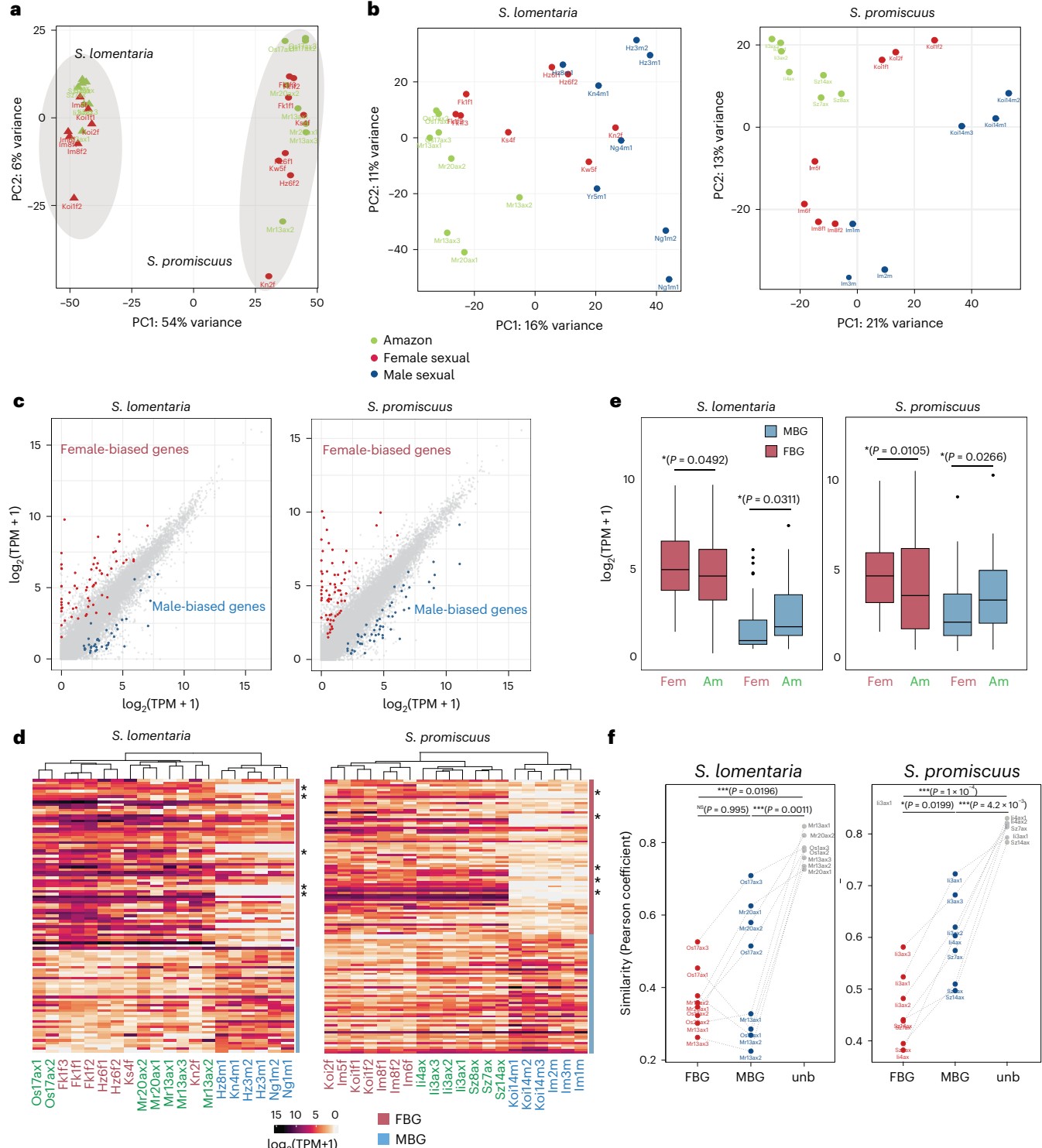

**Fig. 3 | Defeminization and masculinization of gene expression associated with asexuality. a**, PCA plot of female and Amazon RNA-seq samples for both *S. promiscuus* and *S. lomentaria*, using SCOs. **b**, PCA based on all expressed genes per species. **c**, Comparison of gene expression levels, in log₂(TPM + 1), between males and females in *S. lomentaria* and *S. promiscuus*. Dark pink points represent female-biased genes, blue points male-biased genes. Grey points represent unbiased genes. **d**, Hierarchical clustering and heat map of gene expression (in log₂(TPM + 1)) for all the SBG in each species (ComplexHeatmap package, R). The dendogram was constructed using hierarchical clustering based on Euclidean distances (defaults parameters in ComplexHeatmap). Asterisks indicate genes that are female-linked (inside the U-SDR), therefore absent from male samples. Sexual females, males and Amazons are marked in red, blue and green, respectively. **e**, Comparison of gene expression levels in log₂(TPM + 1), using SCO gene sets, during transition to asexuality in *S. lomentaria* and *S. promiscuus*. The boxes represent the interquartile ranges (25th and 75th percentiles) of the data, the lines inside the boxes represent the medians and the whiskers represent the largest/smallest values within 1.5× the interquartile range above and below the 75th and 25th percentiles, respectively. The statistical tests are two-sided Mann–Whitney ranked tests, with *P* values displayed in parenthesis. **f**, Comparison of similarity index values (Pearson coefficients) between expression profiles (in log₂(TPM + 1)) of Amazon and sexual females for unbiased, female (FBG) and male-biased genes (MBG). Significant differences of coefficients between sex-biased genes are indicated directly on the plot (two-sided Mann–Whitney ranked tests); NS, not significant.

words, the transition from female to asexual involves major changes in transcription, mainly at the level of sex-biased genes and mostly so in female-biased genes.

## Conservation of the transcriptomic changes associated with the switch to asexuality

In animals, the events that result in emergence of asexuality from sexual ancestors involve convergent changes in gene expression, that is, in each independent transition to asexuality, the same genes show similar changes in their pattern of expression[9,10]. We thus analysed whether the decay in pheromone production that we observed in the various Amazon populations was accompanied by convergent modifications of their transcriptomes upon each transition to asexuality. The analysis compared the set of SCOs common to *S. lomentaria* and *S. promiscuus* (Supplementary Table 11). Of all expressed SCOs, 6.7% (that is, 320 of the 4,776) exhibited convergent expression, that is, they either upregulated or downregulated their expression in the same way in each of the independent transitions to asexuality (Supplementary Tables 11 and 13; see, for example, the SCO in Fig. 4a). This proportion was significantly greater than what would be expected by chance (determined by using permutation tests, $P = 0.0419$, 10,000 permutations). Functional analysis of these convergently expressed genes identified functions related to oxidative metabolism and ion transport (Fig. 4b and Supplementary Table 14). These convergent genes displayed no differences in their evolutionary rates compared to non-convergent genes (mean dN/dS = 0.195 in genes with convergent expression shifts; mean dN/dS = 0.191 in genes without convergent expression shifts; $P = 0.568$, paired Mann−Whitney *U*-test). Thus, the transition to asexuality involves a small but significant number of genes that consistently change their expression patterns. Yet, these genes show no difference in their evolutionary rates, suggesting that their sequences are not under selective pressure.

The lack of pheromone production in Amazons appears to be fixed in these populations, consistent with previous work suggesting that the lack of pheromone production is inherited in F1 hybrid females of an Amazon and a male in *S. lomentaria*[31]. This raises the possibility that the female sex-determining region of the U chromosome (U-SDR) is involved in the pheromone pathway. If so, we might expect to find differences in expression of U-SDR genes in females and Amazons of both *S. lomentaria* and *S. promiscuus*, where transition to asexuality occurred independently. SDR genes have been described for *S. lomentaria* on the basis of their homology with those of the brown alga *Ectocarpus*[33]; we verified that these genes were also specific to the genome of female *S. promiscuus* and that they were expressed specifically in females.

We compared the expression levels of U-SDR genes in females and Amazons. When all U-SDR genes were analysed together, we found no significant difference in their expression between females and Amazons of either species (Extended Data Fig. 6). Nonetheless, a few U-linked genes were consistently up- or downregulated in the Amazons of both species (Fig. 4c). Notably, a U-specific gene predicted to encode a transmembrane protein containing a Patched domain, similar to the Patched membrane receptor of the Hedgehog signalling pathway, was significantly downregulated in Amazon populations of *S. lomentaria* when compared with females and in at least one of the Amazon lineages of *S. promiscuus* (Supplementary Table 11 and Extended Data Fig. 6). This gene is part of the ancestral group of genes that are thought to have been present in the ancestral female sex locus of brown algae[33].

Together, these data indicate that whereas expression of most U-SDR genes was similar in Amazons and females, certain genes in this genomic region were differentially expressed in both *Scytosiphon* species, consistent with the transition to asexuality.

## Reproductive mode and selection efficacy

Loss of recombination in asexual organisms is predicted to result in changes in the number of neutral polymorphisms segregating in the population and reduced effectiveness of selection (reviewed in ref. 37). Accordingly, there was a significant difference in synonymous diversity ($\pi_S$) between sexual and asexual populations in both species (Fig. 5a). To investigate whether the Amazon populations of *Scytosiphon* species had fewer polymorphisms than sexual populations, as has been shown in asexual diploid organisms[6,38,39], we compared the proportions of genes containing single nucleotide polymorphisms (SNPs). We found a greater proportion of genes containing SNPs in sexual populations of both *S. promiscuus* ($P < 2.2 \times 10^{-16}$) and *S. lomentaria* ($P = 6.32 \times 10^{-16}$) when compared with Amazons. We used a generalized linear model with binomial distribution to analyse the proportion of variable sites among genes that contained SNPs. Sexual populations had a greater proportion of variable sites than had Amazon populations, both in *S. promiscuus* ($P < 2.2 \times 10^{-16}$) and in *S. lomentaria* ($P < 2.2 \times 10^{-16}$) (Extended Data Fig. 7a,b). Demographic modelling using DILS[40] (Methods) confirmed that effective population sizes were reduced in Amazon lineages by a factor 2 in *S. lomentaria* and by a factor 20 or more in *S. promiscuus* (Supplementary Table 3).

We next tested the prediction that the absence of sex and recombination in Amazon lineages results in reduced effectiveness of selection. Purifying selection removes deleterious mutations from a population, thus decreased purifying selection results in accumulation of non-synonymous mutations in protein-coding genes, which are mostly deleterious. The ratio of non-synonymous to synonymous diversity ($\pi_N/\pi_S$) is predicted to be higher in asexual species when compared with sexual species[41]. Note that because gene expression is likely to affect $\pi_N/\pi_S$ (ref. 42), we included it (in $\log_2(\text{TPM} + 1)$) in our models as a dependent variable. Also, to infer the effect of asexuality on $\pi_N$, we conducted likelihood ratio tests with a reduced model without reproductive mode (sexual, asexual). Intriguingly, we did not find the expected higher mean values of $\pi_N/\pi_S$ in Amazon populations of *S. lomentaria* (GLMM $P = 0.713$), and the mean $\pi_N/\pi_S$ in Amazon populations of *S. promiscuus* was actually lower than that of sexual populations (GLMM, $P = 3.47 \times 10^{-15}$) (Fig. 5b). These results would thus suggest that selection is not less efficient in Amazon populations compared with the sexual counterparts.

We also performed the same calculations for the subset of genes in each species that belong to SCOs, as we expect these long-time conserved genes to be maintained under strong purifying selection. The trends of $\pi_N/\pi_S$ in SCOs were similar to those inferred from all genes within species, that is, the mean value of $\pi_N/\pi_S$ in Amazon lineages was either no different (in *S. lomentaria*) or lower (in *S. promiscuus*) than that of sexuals (Supplementary Table 14).

Considering that Amazons spend their life cycle as haploids, the relatively low $\pi_N/\pi_S$ found in Amazons may be caused by haploid purifying selection, which would counteract the negative effects of lack of sex and recombination. We therefore examined whether genes expressed in the haploid stage in sexual individuals display low $\pi_N/\pi_S$ compared with non-specific genes (that is, genes also expressed during the diploid stage). We identified 124 haploid-specific genes in *S. promiscuus* (Supplementary Table 8). In sexual populations, these genes had a mean $\pi_N/\pi_S$ of 0.190, which is significantly lower than that of the non-specific genes (mean $\pi_N/\pi_S = 0.333$) (Mann−Whitney ranked test $P = 2.47 \times 10^{-3}$) (Extended Data Fig. 8). Therefore, considering the fully haploid life cycle of Amazons, their lower $\pi_N/\pi_S$ may be due to haploid purifying selection overwhelming the relaxation of selection due to asexuality.

In *S. lomentaria*, we identified 1,751 haploid-specific genes (Supplementary Table 7). In contrast to *S. promiscuus*, these genes displayed slightly higher $\pi_N/\pi_S$ compared with non-specific genes (mean $\pi_N/\pi_S = 0.416$ and 0.300 for haploid-specific and non-specific genes, respectively; Mann−Whitney ranked test $P = 1.35 \times 10^{-14}$) (Supplementary Table 14 and Extended Data Fig. 7), suggesting that they are less impacted by haploid selection. Thus, it is possible that the fully haploid life cycle in *S. lomentaria* Amazons may not lead to as efficient selection

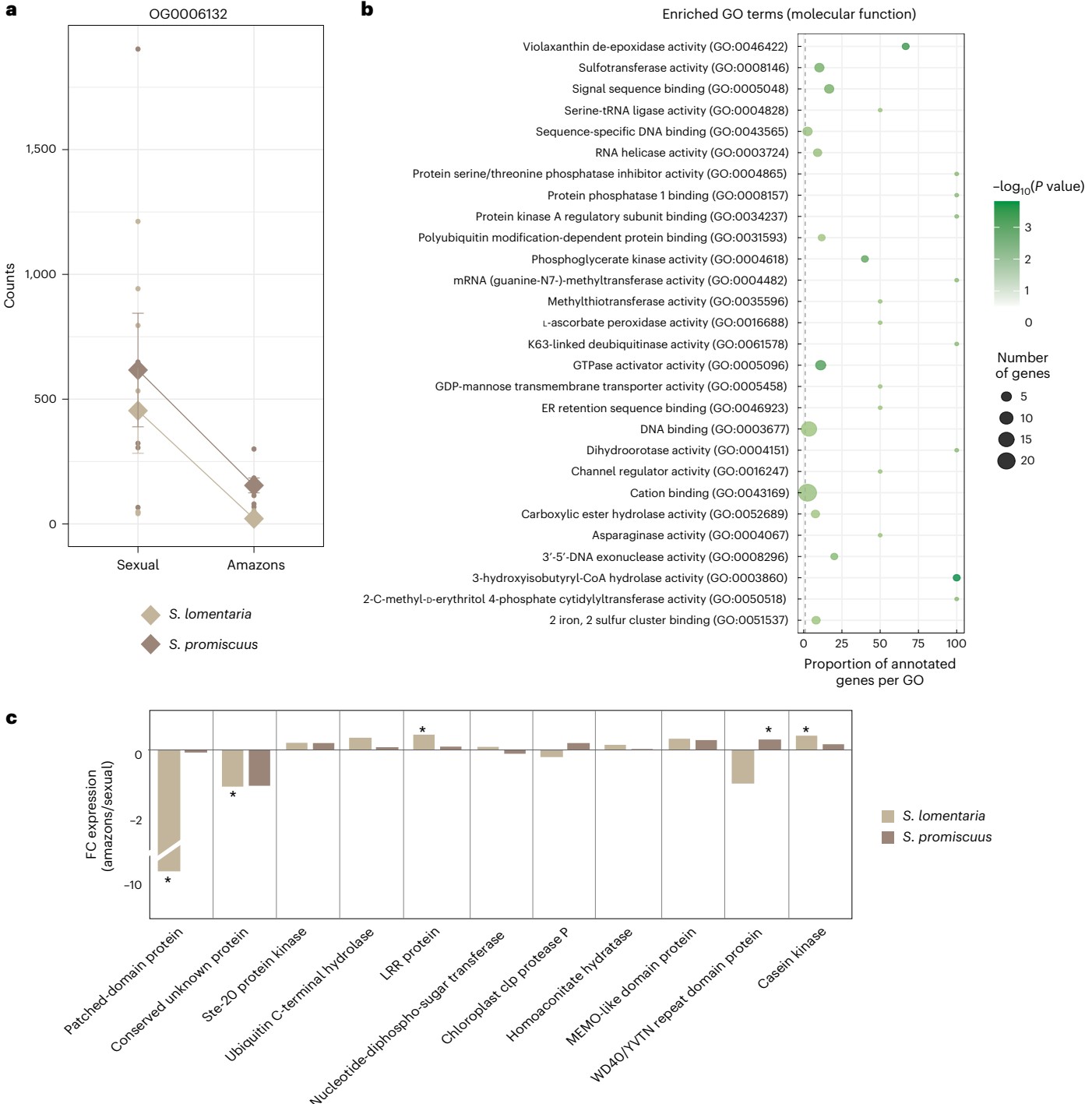

**Fig. 4 | Convergent gene expression changes during transition to asexuality. a**, Example of a single-copy orthologue (OG0006132) showing convergent gene expression changes in both species during transition to asexuality. The plot of normalized counts was generated using the PlotCount function in the Deseq2 package. See also Supplementary Table 10 for details of expression of each SCO showing convergent expression. **b**, GO enrichment (molecular function) in genes that exhibit convergent expression changes during the transition to asexual reproduction. Note that terms related to cellular process and cellular component are presented in Supplementary Table 9. **c**, Changes in expression level (FC) of female U-linked genes in Amazon (asexual) populations compared to ancestral sexual populations. Asterisks indicate significant differences (Wilcoxon test, $P$ = 0.0135, 0.0405, 0.007, 0.0239 and 0.0303, respectively).

as in *S. promiscuus*, in agreement with the absence of difference in $\pi_N/\pi_S$ between Amazons and sexuals.

### An EF-hand-domain-containing protein associated with the Amazon phenotype
To investigate whether any changes at the genomic level were consistently associated with the transition to asexuality in *S. lomentaria*

and *S. promiscuus* females, we searched for genetic variants associated specifically with asexuality in the transcriptomes of females and Amazons. In *S. promiscuus*, we identified 16 transcripts containing non-synonymous mutations that were fully associated with the Amazon phenotype (Supplementary Table 14). Remarkable among these 16 variants, one had a missense mutation that was also fully associated with the Amazon phenotype in *S. lomentaria*. This transcript encodes

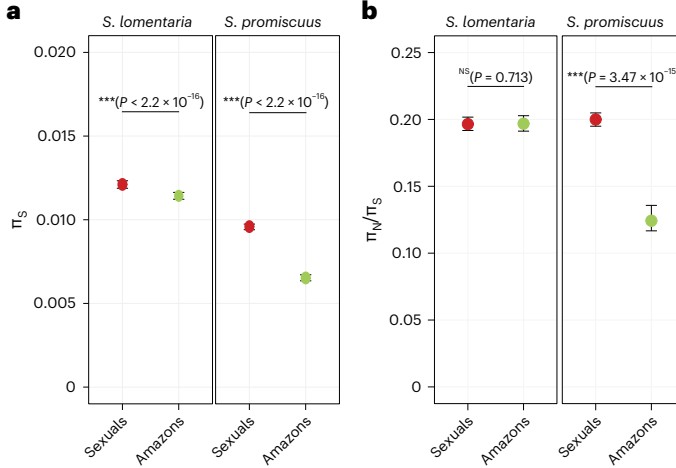

**Fig. 5 | Selection efficacy in sexual versus Amazon populations of**
**_S. lomentaria_ and _S. promiscuus_. a**, $\pi_S$ in sexual versus Amazon populations.
**b**, $\pi_N/\pi_S$ (representing the $\pi_N$ and $\pi_S$ per site) in sexual versus Amazon
populations. Significant differences are indicated inside brackets. Statistical
tests are likelihood ratio tests between linear mixed models including or
excluding the sexual system as independent variable (Methods).

an EF-hand-containing protein. In _S. lomentaria_, the mutation results in
replacement of a glycine residue by a tryptophan (Gly511Trp), whereas
in _S. promiscuus_ it results in replacement of a threonine residue by an
arginine (Thr516Arg) (Fig. 6a and Supplementary Table 15). The muta-
tions occurred in the same exon in both species and correspond to a
similar position in the primary sequence. Importantly, on the basis of
the positions of these species on the phylogenetic tree (see Fig. 1b)
and the fact that these are two independent alleles, we conclude that
these mutations have arisen independently. Note, however, that we
cannot exclude the possibility that this gene is under relaxed selection
following the arrest of pheromone production (or relaxed selection on
genes in other pathways related to sexual traits).

We used molecular modelling to predict the structures of these
proteins and the effects of the mutations on protein structure and
function. The primary structures of the two proteins are highly con-
served between the two species (87.2% identity); they contain three
_Armadillo_-like repeat (ARM-like) domains at their N-termini and two
EF-hand domains at their C-termini (Fig. 6a). ARM domains are typically
involved in protein–protein interactions and intracellular signalling.
Predictions of the folded, three-dimensional structures of the wild-type
(sexual) and variant (Amazon) forms of each orthologue made by
using AlphaFold2 indicate that the structures of the EF-hand domains
are little affected by the mutations (Fig. 6b), thus the calcium-binding
functions of both Amazon proteins probably remain functional, at
least to some extent (Fig. 6b). The additional alpha helix induced by
the mutations in the variants of the Amazons is expected to lead to
altered protein–protein interactions.

## Discussion

Here we studied field populations of two species of the brown alga
_Scytosiphon_ to determine the phenotypic and molecular modifications
that are associated with the transitions to asexuality in organisms with
a haploid-diploid life cycle, where females and males are multicellular
haploid organisms.

### Emergence of obligate asexuality is associated with loss of
### pheromone production and optimization of parthenogenesis

Asexual populations, which reproduce exclusively via female gam-
etes undergoing parthenogenesis, have emerged repeatedly from
sexual ancestors and appear to be frequent in the genus _Scytosiphon_.

Female-specific pheromone production is lost consistently during the
switch to asexuality and this loss is likely to be fixed in these popula-
tions because F1 females have been shown to be incapable of pheromone
production[31]. The fact that F1 females do not produce pheromone is
consistent with a major pheromone locus being inside the U-SDR, that
is, transmitted to all daughters. Substantial further work will be needed
to identify precisely the complex genetic basis of pheromone produc-
tion. Interestingly, the loss of pheromone production is also conspicu-
ous in animals during their transition to obligate parthenogenesis[10],
supporting the idea that this is a costly trait that is rapidly dispensable
in the absence of males.

In all but one population of Amazon we examined, the gametes
were still able to recognize and fuse with male gametes of the same
species, despite having lost the capacity to attract male gametes by
producing pheromone. This observation suggests that the transitions
to asexuality, which occurred <2 Ma, have not led to full loss of sexual
capacity in terms of gamete–gamete recognition and fusion. Note
that given the sex ratio of 0:1 (male:female) in natural populations, it
is unlikely that occasional sex occurs.

Concurrent with the loss of pheromone production, Amazon
female gametes were often larger than the sexual female gametes.
This suggests that after the evolution of obligatory parthenogen-
esis, the female gamete became larger to compensate for the lack of
resources provided by the male gamete. In addition, Amazon female
gametes rapidly engaged in parthenogenetic development. Together,
these observations indicate that Amazon female gametes are special-
ized for asexual reproduction by parthenogenetic development of
unfertilized gametes. In the closely related brown alga _Ectocarpus_
which also has a haplo-diplontic life cycle, gamete parthenogenesis is
controlled by a major quantitative trait locus located in the U-SDR. It
has been suggested that this trait may be subject to both sexual selec-
tion and generation/ploidally antagonistic selection[28]. In other words,
parthenogenesis may be advantageous in situations where there is no
pheromone or when males are absent, whereas producing the phero-
mone is costly and is only advantageous in the presence of males. If
parthenogenesis is not costly, then it can be maintained more easily
than pheromone production. This mechanism would be consistent
with a trade-off[43,44] between the haploid and diploid stages of the life
cycle, where distinct parthenogenesis alleles have opposing effects
on sexual and asexual reproduction and may help maintain genetic
variation[28]. In the case of Amazon populations, the allele that confers
fast and efficient parthenogenesis would be favoured and would rap-
idly spread in the population because it would ensure higher survival
in situations where gamete encounter is limiting.

### Origin of Amazon populations

In animals and land plants, interspecific hybridization and polyploidy
are major triggers for parthenogenesis[2,3,45]. In the brown alga _S. lomen-
taria_, however, phylogenetic analyses based on mitochondrial and
nuclear markers have found no evidence for interspecific hybridiza-
tion[32]. At least two scenarios are possible to explain the origin of the
Amazon populations. In the first scenario, a mutation leading to loss of
pheromone production triggers the loss of sex (that is, 'spontaneous
origin')[3]. Because female gametes are capable of parthenogenesis, then
the female parthenogenetic population expands, whereas males would
decay because they cannot undergo efficient parthenogenesis[46]. In the
second scenario, males are less tolerant to a change in environmental
conditions (for example, water temperature)[32,46], so they become less
frequent, which favours females that rapidly reproduce by parthe-
nogenesis. As pheromone production is presumably costly, this trait
would then be lost. In this scenario the loss of pheromone is secondary
to the loss of sex.

Our results highlight similarities between brown algae and ani-
mals, despite the differences in sexual systems and the large evolu-
tionary distance. In both, obligate asexuality evolves from already

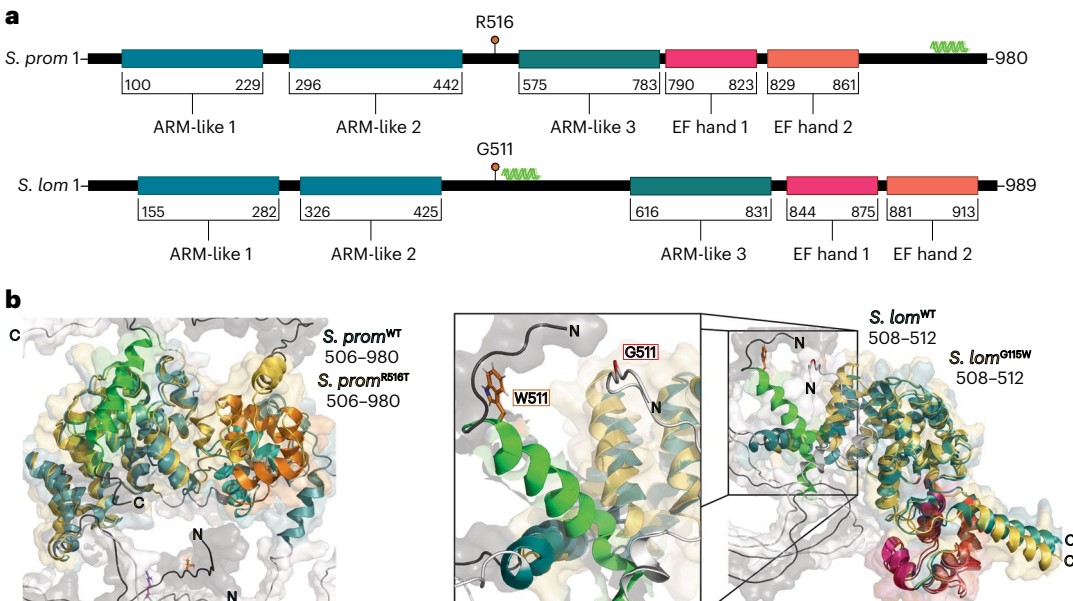

**Fig. 6 | Domain architecture and models of EF-hand–ARM homologues from *S. lomentaria* and *S. promiscuus*. a**, Protein domain architecture of both protein-coding genes from *S. lomentaria* and *S. promiscuus*. Both orthologues contain three ARM-like repeat domains (dark blue) that encapsulate large, unfolded loops, as well as two C-terminal EF-hand motifs (magenta/orange). Both variant sites are highlighted (red lollipop) and precede the third ARM-like domain. The additional α-helix predicted with each mutant protein is highlighted (light green). **b**, Generated models of the wild-type (sexual) and variant proteins from each species. The Thr516Arg substitution is distal to the EF-hand motifs (left) and are predicted to have little impact on structural integrity. While both variants have an additional α-helix that is within proximity to the C-terminal ARM domains, only the Gly115Trp mutation within the *S. lomentaria* variant initiates the newly acquired α-helix (right) following the mutation site.

existing pathways (that is, 'facultative' parthenogenesis)[2], a situation that may be rather different from evolving an entirely novel asexual pathway. Indeed, parthenogenesis capacity is already present in female *Scytosiphon*, hence the spore-production pathway specific to parthenosporophytes was already present at the time of the transition to obligate asexuality, and this may have facilitated the transition to a fully, obligatory asexual life cycle.

Interestingly, our Amazon populations did not undergo the mutational meltdown that is predicted to arise in parthenogenetic populations due to the absence of recombination. In other words, selection is not less efficient in Amazon populations than in their sexual counterparts. Amazons spend their entire life cycle in the haploid stage (see Fig. 1). Therefore the lack of mutational meltdown in Amazons may be due to haploid purifying selection overwhelming the relaxation of selection due to asexuality and avoiding the deleterious consequences of prolonged asexuality observed in animal systems with diploid life cycles[6,39,47]. Our results from *S. promiscuus* populations thus support the idea that haploid purifying selection overwhelms the effect of asexuality and efficiently purges deleterious mutations[48,49]. Note, however, that *S. lomentaria* Amazons appear to be less efficient in overcoming the consequences of lack of sex. Why haploid selection differs between the two species and how asexuality and haploidy interact remain open questions. Note that additional factors such as different age of asexuality or different magnitude of gene flow between sexuals and asexuals may contribute to the different patterns observed between the two species.

### Defeminization of gene expression driven by decay of female traits?

Males and females have distinct phenotypes, despite sharing most of their genome. A possible resolution of this apparent conflict is by differential gene expression, whereby genes are expressed at different levels in each sex[50]. This differential gene expression, however, may lead to conflicts between the optimal expression of genes in males and females. Our data allowed us to test the hypothesis that gene expression in (haploid) females is constrained from evolving to its optimum level due to sexually antagonistic selection on males, by examining changes in sex-biased gene expression in the Amazon populations, which do not produce males. We expected the transcriptome of Amazons to be feminized, as they no longer experience sexual conflict, thus gene expression could reach its female optimum. However, as in animals[9,10], gene expression was defeminized, simultaneous with loss of the female-specific trait of pheromone production. We propose that the changes in sex-biased gene expression during the switch to asexuality in this organism are thus mainly related to the loss of (costly) female traits and consequent changes in trait optima and not necessarily to resolution of sexual conflict. Since sexually dimorphic traits are thought to be a product largely of sex-biased gene expression[51], a link between reduced female sexual traits and reduced female-biased gene expression is a plausible explanation for the decreased expression of female-biased genes we observe and the dramatic change in female-biased pattern of gene expression (low Pearson similarity index). Note however that in *S. lomentaria*, these results may be partially conflated by regression to the mean effects (Extended Data Fig. 4). Transcriptome defeminization in *S. lomentaria*, if present, is probably weaker than in *S. promiscuus*, which is in agreement with the more limited phenotypic changes in *S. lomentaria*.

Finally, we also note the complexity of the Amazon phenotype, in that some traits are more male-like (absence of pheromone), corresponding to 'masculinization', whereas others are more pronounced female traits (large gamete size), that is, corresponding to 'feminization'. Our results indicate that, for gene expression, the former category outweighs the latter, perhaps reflecting a degeneration of costly traits mediating male–female interactions in sexual populations.

### Transcriptome convergence and haploid selection in Amazons
The independent, convergent decay in pheromone production that we observed in the various Amazon populations was accompanied by

convergent modifications of their transcriptomes upon each transition to asexuality. The transcriptomes of Amazons were more closely related to each other than to those of their phylogenetically closest sexual population, suggesting that similar regulatory networks are recruited repeatedly and independently during the transition to asexuality. The convergent gene expression changes that were associated with the transition to asexuality did not, however, appear to be driven by selection. It is possible that rapid engagement in parthenogenesis to develop the parthenosporophyte generation (Fig. 1) explains the convergence in transcriptomes. Interestingly, convergently recruited genes in Amazon populations have putative functions similar to those enriched in the sporophyte generation of *S. lomentaria*[52]. Considering that Amazons are optimized for parthenogenesis and rapid parthenosporophyte development (see above), it is therefore likely that their changes in gene expression reflect a switch in life cycle strategy.

The U-specific region harbours several genes that converge in terms of their changes of expression during the switch to asexuality. While phenotypic sexual dimorphism is not necessarily caused by genes within the non-recombining sex-determining region, genes within the U-SDR could eventually be involved in the phenotypic changes that underlie the transition to asexual reproduction. Interestingly, a major locus controlling parthenogenesis is linked to the SDR in a closely related brown alga (*Ectocarpus* sp.) and it is likely that pheromone production may also be controlled by the sex locus. It is impossible, however, to determine whether the convergent changes in expression of SDR genes are a cause or effect of the asexuality; SDR genes that are differentially regulated in Amazons may be involved in triggering asexuality or may simply be a consequence of this developmental switch.

The Amazon phenotype was genetically associated with a change in the protein-coding sequence of a gene encoding a putative *Armadillo*- and EF-hand-domain-containing protein. Reverse genetic tools are not yet available for *Scytosiphon*, therefore a causal relationship cannot currently be established between this protein and asexuality. $Ca^{2+}$ plays an important role in signalling pathways in brown algae[53,54] and in pheromone signalling pathways in other eukaryotes[55], and *Armadillo* domains have well-characterized roles in protein–protein interactions and membrane binding[56]. It is possible that a membrane binding ability of the Amazon variant protein is compromised, or a protein–protein interaction that is essential for $Ca^{2+}$-dependent pathways during sexual reproduction and/or pheromone production is disrupted.

## Methods

### Phenotyping

We focused on four gamete phenotypic traits: fertilization rate when crossed with gametes of the opposite sex, sex pheromone production, gamete size and capacity for parthenogenetic development. These phenotypic traits were examined in female-only, asexual populations, Ii and Sz, and they were contrasted with the closest related sexual (ancestral) populations (that is, Ii was compared with Im, and Sz was compared with Koi) (Supplementary Table 1). Culture strains were maintained using plastic Petri dishes (90 × 20 mm) and PESI medium[57] at 15 and 10 °C in long-day (16 h:8 h, light:dark) and short-day (8 h:16 h, light:dark) conditions with 30–50 µmol m$^{-2}$ s$^{-1}$ photon flux density.

To examine fertilization rate of gametes from parthenogenetic populations, they were crossed with male gametes from sexual populations. The crossing was performed following ref. 31. We added in large excess male gametes to female gametes in a small drop that settled in a microscopic field and observed the proportion of the female gametes that formed zygotes in 5 min. As control, crossing between female and male gametes from sexual populations was also performed. For the statistical analysis of the fertilization rate (the number of fertilized gametes and unfertilized gametes), a GLMM model was adopted with a binomial distribution. Models with and without sexuality

(Amazon, sexual female and sexual male) were established for the fertilization rate (identity of culture strains was considered as the random effect in both models), and the Akaike information criterion (AIC) values and parameters for each were examined (Supplementary Table 3). The modelling and model selection were conducted in R using the package lme4 (ref. 58).

Brown algae sex pheromones are hydrophobic, cycloaliphatic unsaturated hydrocarbons comprising 8–11 carbon atoms[34]. Sexual pheromone was detected using GC–MS following ref. 32. Fertile gametophytes were kept in 300 ml of sterilized seawater in a 500 ml flask at 15 °C and gametes were released in the flask. Volatile secretions were trapped on Mono Trap RCC18 (GL Science) using a closed-loop-stripping system at 15 °C. After looping for 12 h, absorbed volatile compounds were eluted with 50 µl $CH_2Cl_2$ and immediately analysed by GC–MS using Zebron ZB-wax columns (Phenomenex, 30 m × 0.25 µm; He as the carrier gas; programme rate: 45–200 °C at a rate of 5 °C min$^{-1}$). Compounds were identified using the NIST MS library and similarity search programme[59]. Females and males from sexual populations were also analysed as positive and negative controls, respectively. Sex pheromone was also detected by olfaction in blind tests.

Several minutes after gamete release, gametes lose their motility and settle to the substratum and then change their shape from pear-shaped to spherical. We took images of spherical gametes with a Nikon Digital Sight DS-Fi1 and measured their diameter using ImageJ software[60]. Sixteen to 50 gametes were measured for each individual strain. For the statistical analysis of the gamete size, a GLMM was adopted with a normal distribution.

To examine parthenogenetic development, gametes were cultivated at 15 °C in long-day conditions as described above. We recoded the number of germinated gametes and non-germinated gametes after 24 h cultivation, as well as the number of cells of each germling after 5-day cultivation. For each individual, between 40–160 gametes/germlings were scored. For the statistical analysis of the germination rate and cell numbers of germlings, a GLMM was adopted with a binominal distribution and a Poisson distribution, respectively (Supplementary Table 3). All statistical analyses were performed in R.

### Genome sequencing and assembly

Genomes of *S. lomentaria* (strain Kn2f) and *S. promiscuus* (strain Ii3ax and Im6f) were sequenced to be used as references for analyses of RNA-seq data. Before the genome DNA extraction, the gametophytes were cultivated with antibiotics (Penicillin G, 50 mg l$^{-1}$; ampicillin, 50 mg l$^{-1}$; chloramphenicol, 5 mg l$^{-1}$; and kanamycin, 0.5 g l$^{-1}$) for 1 week to remove eventual bacterial contamination. The gametophytes were frozen in liquid nitrogen and crushed to powder using TissueLyser II (QIAGEN), and genomic DNA was extracted using OmniPrep for Plant (GBiosciences) as in refs. 18,61. The libraries of Kn2f and Im6f were sequenced on an Illumina Nextseq 2000 system with paired-end reads of 150 bp. *S. promiscuus* strain Ii3ax was sequenced using Oxford Nanopore Sequencing (ONT) MinION. OmniPrep and NucleoBond for high molecular weight DNA were used to extract the genomic DNA. ONT sequencing was performed on an R9.4.1 flow cell with the SQK-LSK110 ligation sequencing kit.

The *S. promiscuus* nanopore ONT reads were basecalled by the ONT basecaller Guppy v.6.3.8 + d9e0f64 and the configuration file dna_r9.4.1_450bps_sup.cfg. The fastq reads were filtered for bacterial contamination by using a conjunction of Kraken2 (v.2.1.2)[62] and blastn (v.2.9.0+)[63] in combination with the NCBI nt database (download date 1 July 2022). The filtered reads were assembled using SMARTdenovo (options J 500, k 32)[64]. Super scaffolding was performed using ONT medaka_consensus v.1.7.2 (https://github.com/nanoporetech/medaka) with default options. Gap closing was performed in PBJelly v.v15.8.24 (https://github.com/esrice/PBJelly)[65] using *S. promiscuus* strain Im6f Illumina read contigs.

For *S. lomentaria*, the quality of the scaffolded genomes was assessed using QUAST (v.5.0.2)[66], and statistics are shown in Supplementary Table 1.

## Genome structural annotation

All genomes were soft masked using Repeatmasker (v.4.1.2) after building a de novo transposable elements and repeats database with RepeatModeler (v.2.0.3)[67]. Two runs of BRAKER2 (v.2.1.6)[68] were used to predict gene sets employed for all downstream analyses: (1) using input predicted protein from model species *Ectocarpus* sp. (EctsiV2_prot_LATEST.tfa)[69,70] and *Chordaria linearis* (Ectocarpales[18]) and (2) using data from the same individual sequence mapped onto the genome with tophat2 (v.2.1.1 (ref. 71)). TSEBRA[72] was then used to select transcripts from the previous two runs of BRAKER2 with parameters recommended to include ab-initio predicted genes. Genome annotation completeness was assessed using BUSCO (v.3)[73] against the eukaryote and stramenopile gene set Odb10[74].

## RNA sequencing

Gametophytes were cultivated in plastic Petri dishes (90 × 20 mm) and sterile natural seawater from the North Sea enriched with full strength PESI medium at 10 °C in long-day (16 h:8 h, light:dark) conditions with LED lighting of 20 µmol m$^{-2}$ s$^{-1}$ photon flux density. Medium was renewed every week until gamete collection.

RNA libraries from at least three replicate individuals per sex for each lineage (Supplementary Table 1) were prepared from gametes using NEBNext Single Cell/Low Input RNA Library Prep Kit for Illumina (New England Biolabs). Mature gametophytes release gametes just after medium renewal. Gametes were collected by phototaxis: *Scytosiphon* gametes have negative phototaxis, so freshly released gametes accumulate on the opposite side of a light source in a Petri dish. Between 300–10,500 gametes in 0.5 µl of medium were transferred to a PCR tube and incubated for 5 min in darkness at 10 °C to allow gametes to settle on the surface of the tube. After incubation, 5 µl of the NEBNext cell lysis buffer was added to the tube, immediately followed by reverse transcription and library preparation according to manufacturer protocol. Libraries were sequenced on an Illumina Nextseq 2000 system with paired-end reads of 150 bp.

## Phylogenetic tree construction based on RNA-seq data

To examine phylogenetic relationships among parthenogenetic and sexual lineages, phylogenetic trees were generated on the basis of assembled RNA-seq data. Adapter sequences and low-quality reads were removed using Trimmomatic[75]. The filtered reads were mapped on either *S. lomentaria* or *S. promiscuus* genome using Tophat2 (v.2.1.1). The resulting bam files were sorted and used for genome-guided transcriptome assembly using Trinity (v.2.13.2)[76]. Coding sequences (CDS) were predicted for the longest isoform of each assembled transcriptome using Transdecoder (v.5.7.0). To remove transcriptomes from potential contamination, amino acid sequences from the predicted CDS were blasted against algae_peptide databases (algal sequences downloaded from the NCBI and the JGI databases) using DIAMOND[77] in sensitive mode, and proteins without blast hits were removed. To remove highly identical amino acid sequences, CD-HIT[78,79] was used with the parameter '-c 0.95'. Then, using the filtered datasets, single-copy orthologues among the samples were predicted using OrthoFinder (v.2.5.4)[80]. Seventy-eight single-copy orthologues were detected, and the nucleotide sequences of 53 genes whose missing rate was <20% were concatenated and used for phylogenetic analysis. The phylogenetic analysis was performed with IQ-TREE (v.2.1.4)[81] using flag '-MFP + MERGE' with 1,000 replicates of ultrafast bootstrapping.

## Divergence time estimation

The divergence time between lineages was estimated using *cox1*, *cox3* and *rbcL*[46,82]. We constructed a time-calibrated tree of 23 species of Ectocarpales and Laminariales (Supplementary Table 15) based on the mitochondrial *cox1*, *cox3* and chloroplast *rbcL* (total 2,819 bp) using BEAST (v.2.7.4)[42,83] with the following settings: substitution models of GTR + Γ for *rbcL* and *cox3*, and HKY + Γ for *cox1*, an optimized relaxed clock, Yule Model prior and 131,815,000 generations of Markov chain Monte Carlo (MCMC) with sampling every 1,000 generations. In addition, we specified a prior on the root age of *Nereocystis luetkeana* and *Pelagophycus porra* as a calibration point, similar to ref. 84. We assumed that the two species are monophyletic and the lower boundary of their divergence time is 13 Ma (exponential distribution with mean and offset of 2.0 and 13.0, respectively). Stationarity of the MCMC ran was checked using Tracer (v.1.7.1)[85]. A maximum clade credibility tree with median node heights was constructed using TreeAnnotator[86] with burn-in of 10%. The constructed tree was visualized using FigTree (v.1.4.3). Note that we could not estimate the divergence times of a subset of the samples because there were not enough polymorphisms: in *S. promiscuus*, sexual and Amazon populations (Koi and Sz, and Im and Ii) have identical *cox1* haplotypes, so divergence time cannot be estimated. Regarding *S. lomentaria*, we cannot distinguish sexual from asexuals using *cox1* (ref. 32). The divergence times between *S. lomentaria* and *S. promiscuus* was estimated as 13.7 Ma (median; 95% highest posterior density (HPD)): 8.1–21.9 Ma), and between *S. promiscuus* Koi and *S. promiscuus* Im populations as 0.76 Ma (median; 95% HPD: 0.1–1.8 Ma).

## Population genomic modelling of population divergence

In addition to phylogenetic dating, we used an alternative and independent approach to date the origin of asexual lineages. We used an approximate Bayesian approach as implemented in DILS[40] to reconstruct the history of divergence between pairs of sexual and asexual populations. In brief, DILS modelled two diverging populations from an ancestral one, allowing for change in population sizes and possible gene flow between the two populations. It also takes into account the heterogeneity in effective population size and migration rate that can be generated by selection, and which can bias inferences. We used synonymous SNPs from RNA-seq data and run the analyses on three pairs: *S. lomentaria* sexuals/Amazons, *S. promiscuus* sexuals/Amazons Koi and *S. promiscuus* sexuals/Amazons Im. To calibrate the model, we used a mutation rate of $1.22 \times 10^{-9}$ as estimated experimentally in *Scytosiphon*[87]. The details of the options and priors used are given in Supplementary Table 3.

## Expression quantification and identification of sex-biased genes in sexual populations

RNA-seq reads were removed if the leading or trailing base had a Phred score <3, or if the sliding window Phred score, averaged over four bases, was <15. Reads shorter than 36 bases were discarded (as well as pairs of reads where one of the pairs was <36 bases long). Trimmomatic-processed RNA-seq reads from each library were used to quantify gene expression with kallisto (v.0.46.2)[88] using 31-bp-long *k*-mers and predicted transcript of each species. RNA-seq libraries were all composed of paired-end reads. A gene was considered expressed in a given species and/or a given sex when at least two-thirds of samples displayed an expression level above 0.4 transcripts per million (TPM). Estimates of sex-biased gene expression in dioicous sexual populations were obtained using read count matrices as input for the DESeq2 package[89] in R (3.6.3). *P* values were corrected for multiple testing using the Benjamini–Hochberg's algorithm in DESeq2, applying an adjusted *P* value cut-off of 0.05 for differential expression analysis. Genes with a minimum of 2-fold change in expression level between sexes were retained as sex biased. We plotted heat maps of expression levels in $\log_2(\mathrm{TPM} + 1)$ for each sample within each species using the ComplexHeatmap R package[90].

To further test whether there is bone fide defeminization/masculinization in the Amazons, we compared the similarity in gene

expression using Pearson correlations in Amazons versus males and females versus males, separately per species. In *S. promiscuous*, Amazons are more similar to males than the females are to males, indicating that indeed in this species, there is a clear defeminization/masculinization of gene expression in Amazons, but the result is less clear for *S. lomentaria* (Extended Data Fig. 4c,d). We also took the genes that are male biased in a male versus Amazon comparison and checked their expression in females and Amazons. We found that these are higher expressed in females than in Amazons. Genes that are Amazon biased (in the male versus Amazon comparison) are downregulated in sexual females, indicating that regression to the mean bias cannot be excluded. Together, these tests suggest defeminization and masculinization of sex-biased expression in *S. promiscuus* Amazons, but that the results may be conflated by regression to the mean effects, specifically in *S. lomentaria*.

To study generation-specific gene expression (that is, expression in the haploid versus diploid stages of the life cycle), we used datasets available from ref. 52 for *S. lomentaria*, and we generated RNA-seq data for triplicate samples of male and female haploid gametophytes and triplicate samples of diploid sporophytes of *S. promiscuus* (Supplementary Table 1). RNA-seq analysis was performed as described above. We considered a gene to be gametophyte specific if it was expressed in at least four of the six gametophyte libraries and not expressed in the sporophytes.

### Gene expression at U-specific genes in females versus Amazons

Expression values (TPM) of U-SDR genes in males, females and Amazon populations of *S. lomentaria* and *S. promiscuus* were compared in the context of sex-linkage genes. The *Scytosiphon* sex-linked genes were extracted from ref. 33 on the basis of orthology with *Ectocarpus* genes. If an *Ectocarpus* gene model had hits on multiple *Scytosiphon* gene models (*Scytosiphon* gene models were split or truncated), the *Scytosiphon* gene models were merged to represent a complete *Ectocarpus* model. To identify significant differences in gene expression, a Wilcoxon test was performed for each U-sex-linked gene.

### Orthology and evolutionary sequence divergence between *Scytosiphon* species

We inferred SCOs between the two species using Orthofinder (v.2.5.2) with default parameters[80]. We used kallisto (v.0.46.2) to quantify expression levels of SCOs. Orthologous proteins between species pairs were aligned with MAFFT (v.7.453)[91], and the alignments were curated with Gblocks (v.0.91b)[92] and back translated to nucleotides using translatorX[93]. We used these nucleotide alignments as input for phylogenetic analysis by maximum likelihood (PAML4, CodeML)[94] to infer pairwise dN/dS ($\omega$) with the F3 × 4 model of codon frequencies. We retained orthologues with 0 < dS < 2 as valid for further analysis. We compared species evolutionary rates separately for female-biased, male-biased and unbiased genes using Mann–Whitney ranked test. No orthogroup presented an inconsistent sex bias between the two species, that is, biased in one sex in one species and in the opposite sex in the other species.

### Polymorphism (SNPs) inference within *Scytosiphon* species

RNA-seq data of all strains shown in Fig. 1c were used to detect synonymous and non-synonymous mutations in the context of Amazon-specific mutations (female versus Amazon). *S. promiscuus* and *S. lomentaria* data were treated the same way but kept separate by the two strains. Filtered RNA-seq data from all samples were used to detect variants according to the pipeline of ref. 95. Software updates were accomplished for GATK (v.4.2.6.1)[96] and gmap-gsnap (v.2021-12-17)[97]. Only variants that passed the filter steps (read depth ≥9 and supporting reads ≥7) and are located on exon features were kept for further analysis.

The reference genome of *S. lomentaria* is a female (Kn2f) and the reference genome of *S. promiscuus* is an Amazon (Ii3ax). The filter was set to a read coverage of ≥9 reads, and ≥7 reads have to support the alternative base (according to ref. 95). Non-synonymous variants were detected using SNPEff (v.5.1d)[98] run in default mode. Exclusive variants represent the unique section of the intersection between female, male and Amazon samples. Candidates are variant positions present only in Amazon samples and all Amazon strains have to contain the variant. Since the variant calling may miss some SNPs or InDels, each of the *S. promiscuus* candidate positions were manually checked with JBrowse (if the mapping files (bam) do show exclusive ax SNPs). Twenty-five candidate positions were identified as Amazon specific and used for further analyses. The 25 positions were located on 16 gene models. Fifteen gene models had orthologues on *S. lomentaria*. By manually checking the variant positions on the 15 *S. lomentaria* orthologues, a single specific Amazon variant was identified across both species. The *S. lomentaria* gene is anno1.g11893 (pos. Contig91_RagTag:1006619), and the *S. promiscuus* gene is anno1.g1469 (pos. Contig91:1065707).

### Estimation of $\pi_N/\pi_S$

Variants were filtered to retain SNPs present in at least two individuals per group, with a minimum coverage of 7 reads and a minimum average phred quality of 20. We performed an annotation on the basis of cleaned, filtered vcf files with SNPGenie[99]. Following recommendations, SNPGenie was run twice: once for the forward strand and once for the complementary strand, each time with argument vcfformat =4.

To test for differences between sexual systems, we used generalized linear models with binomial distribution to analyse the proportion of genes that displayed SNPs and the proportion of variable sites in genes that had SNPs. We included the log-transformed expression level (in TPM) as independent variable and the gene identity as a random factor. Models were computed separately for each *Scytosiphon* species. We used linear mixed models to infer the effect of the sexual system on $\pi_S$ and $\pi_N/\pi_S$ separately per species, including gene expression level as an independent variable and gene identity as a random factor. These analyses were repeated on the SCO gene subset.

### Expression changes between sexual and asexual females

We compared the expression levels of female- and male-biased genes in sexual versus asexual females with Mann–Whitney ranked tests to investigate the possible masculinization and defeminization of Amazon expression profiles, separately for each species.

Convergent changes associated with transitions to asexuality were investigated on SCOs inferred across the two species. We used DESeq2 including expression data from sexual and asexual (Amazon) females only (males were removed) to model the gene expression as a function of the species, the sexual system (sexual or asexual) and the interaction between these two variables. We retained convergently differentially expressed genes by performing a likelihood ratio test when they were significantly affected by the sexual system but not by species or by the interaction between species and the sexual system. We further tested whether this observed number of convergent genes was greater than what would be inferred by chance, using a permutation test (10,000 iterations) where the sexual system (sexual versus asexual) of individual gene count matrices was randomized, making sure that replicates of the same population and sex shared the same randomized sexual system. The above-described DESeq2 inference of convergent gene expression was then repeated for each of the 10,000 permuted datasets.

The SuperExactTest package in R was used to perform exact multiset intersection tests to determine whether representation of sex-biased genes among genes showing convergent expression was greater than expected by chance ($P < 0.05$).

To further characterize the role and fate of sex-biased genes during the transition to asexuality, we used an approach similar to that

in ref. 18. We compared the expression profiles of expressed genes (in $\log_2(TPM + 1)$) between sexual and Amazon females by computing the Pearson correlation coefficients of expression levels within species, between the mean expression across all sexual females and each asexual female sample, separately for female-biased, male-biased and non-biased genes. We compared Pearson coefficients of regression within each species using the cocor package[100], considering gene expression profiles of SBGs and unbiased genes within sexes as dependent. We report the *P* value based on refs. 100,101.

Evolutionary sequence divergence (dN/dS) of SCOs with convergent expression shifts was compared to that of the rest of the SCOs to infer whether these genes experienced different selective pressure (Mann–Whitney ranked tests).

### Functional annotation analysis

Predicted genes for each species were blasted against the NCBI non-redundant (nr) protein database using Diamond (v.2.0.15)[77]. Functional annotation was performed using BLAST2GO (ref. 102), as well as the InterProScan (v.5.59-91.0) prediction of putative conserved protein domains[103]. Gene set enrichment analysis (GSEA) was carried out separately for each species and each gene set (female-biased, male-biased), as well as for genes with convergent expression shift with asexuality (combining gene ontology (GO) terms inferred for each species for genes within orthogroups) using Fisher's exact test implemented in the TopGO package, with the weight01 algorithm[104]. We investigated enrichment in terms of molecular function ontology and reported significant GO terms with $P < 0.05$. All statistical analyses were performed in R (4.2.3), and plots were produced using ggplot2 in R[105].

### Structure prediction

Primary sequences were aligned using ClustalW and T-Coffee, followed by JPred secondary structure prediction using Jalview[106–108] to determine the putative domain boundaries. A custom script to predict the protein models with AlphaFold2 was utilized to generate all protein models in this analysis[109]. Only protein models with the highest-ranked predicted local distance difference test (pLDDT) score were selected.

### Reporting summary

Further information on research design is available in the Nature Portfolio Reporting Summary linked to this article.

## Data availability

Accession codes are given in Supplementary Table 1.

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

## Acknowledgements

This work was supported by the MPG, the ERC (grant no. 864038 to S.M.C.), the JSPS Overseas Research Fellowships (to M.H.), the BMBF-funded de.NBI Cloud within the German Network for Bioinformatics Infrastructure (de.NBI) (031A532B, 031A533A, 031A533B, 031A534A, 031A535A, 031A537A, 031A537B, 031A537C, 031A537D, 031A538A). S.M.C. was supported by the Moore Foundation (GBMF11489) and the Bettencourt-Schuller Foundation. We thank M. Hiraoka for help with sampling and phenotyping *S. promiscuus*, E. Avdievich for help with Nanopore sequencing of *S. promiscuus*, D. Roze for helpful discussions and C. Featherstone for assistance in the preparation of the manuscript. V. Alva and J. R. Weir shared the custom AlphaFold2 script that was utilized to generate the protein models.

## Author contributions

M.H. conceptualized the project, conducted investigations, led formal analysis, performed visualization and wrote the original draft. G.C. conceptualized the project, conducted investigations, developed the methodology and performed visualization. F.B.H. supported the conduct of investigations and the development of the methodology, and led data curation. E.I.K. supported the conduct of investigations and the visualization. K.K., T.J. and T.W. supported the conduct of investigations. S.G. supported the conduct of

investigations and developed the methodology. S.M.C. led project conceptualization, funding acquisition, project administration, and review and editing of the manuscrip; developed the methodology and supported visualizaton.

## Funding

## Competing interests

The authors declare no competing interests.

## Additional information

**Extended data** is available for this paper at https://doi.org/10.1038/s41559-024-02490-w.

**Correspondence and requests for materials** should be addressed to Susana M. Coelho.

[1]Department of Algal Development and Evolution, Max Planck Institute for Biology Tübingen, Tübingen, Germany. [2]Department of Biological Sciences, Faculty of Science, Hokkaido University, Sapporo, Japan. [3]Faculty of Pharmaceutical Sciences, Hokkaido University, Sapporo, Japan. [4]Laboratoire ECOBIO (Ecosystèmes, biodiversité, évolution), UMR 6553, CNRS, Université de Rennes, Rennes, France. [5]Department of Ecology and Genetics, Evolutionary Biology Centre, Uppsala University, Uppsala, Sweden. [6]Present address: Research Center for Inland Seas, Kobe University, Rokkodai 1-1, Nadaku, Kobe, Japan. ✉e-mail: susana.coelho@tuebingen.mpg.de

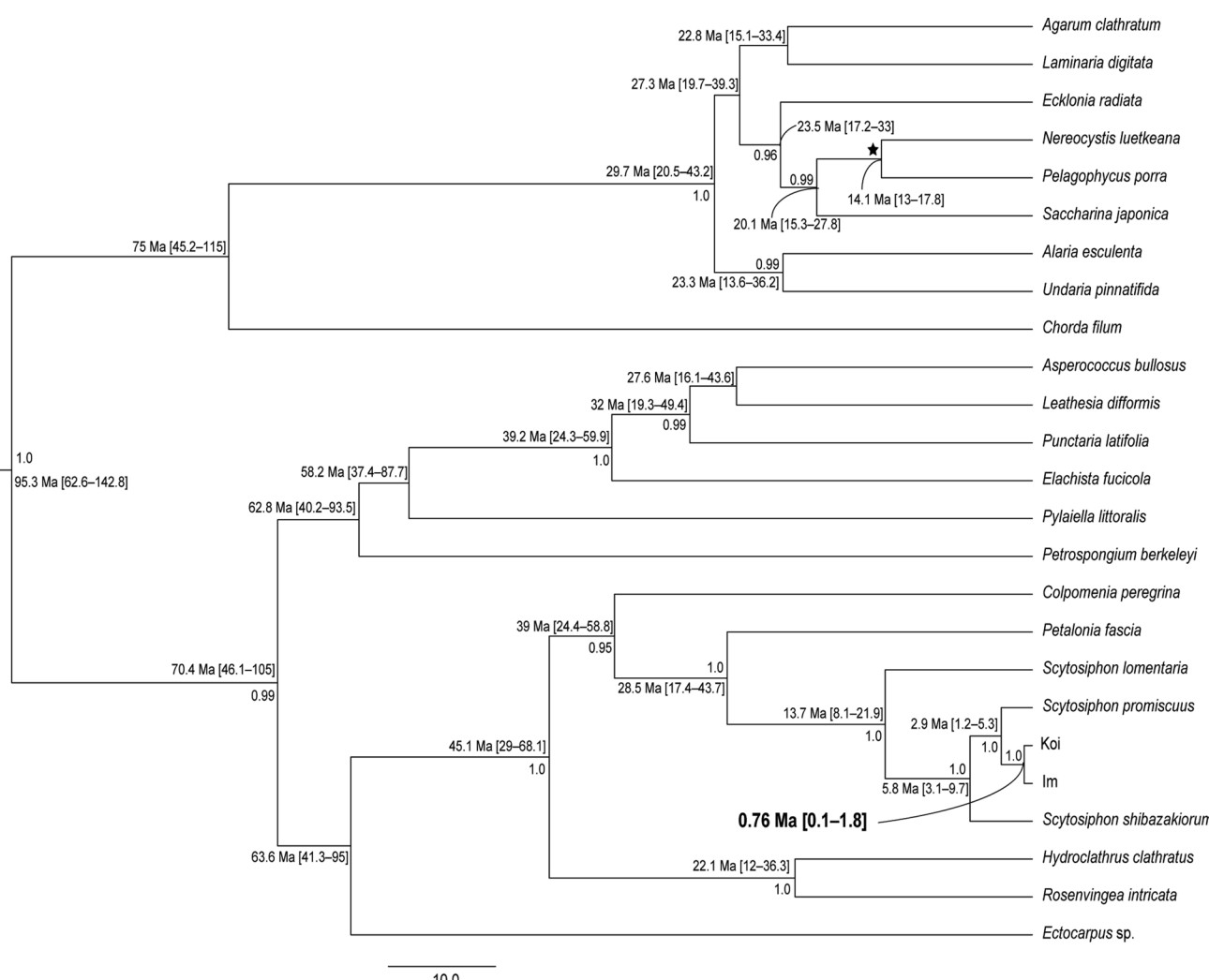

**Extended Data Fig. 1 | Phylogenetic tree showing divergence time.** Posterior probability of > 0.95 and the median value of estimated divergence time (95% highest probability density in square brackets) are given to each node. The black star indicates the node used as a calibration point, an exponential prior with 13 Ma as lower boundary.

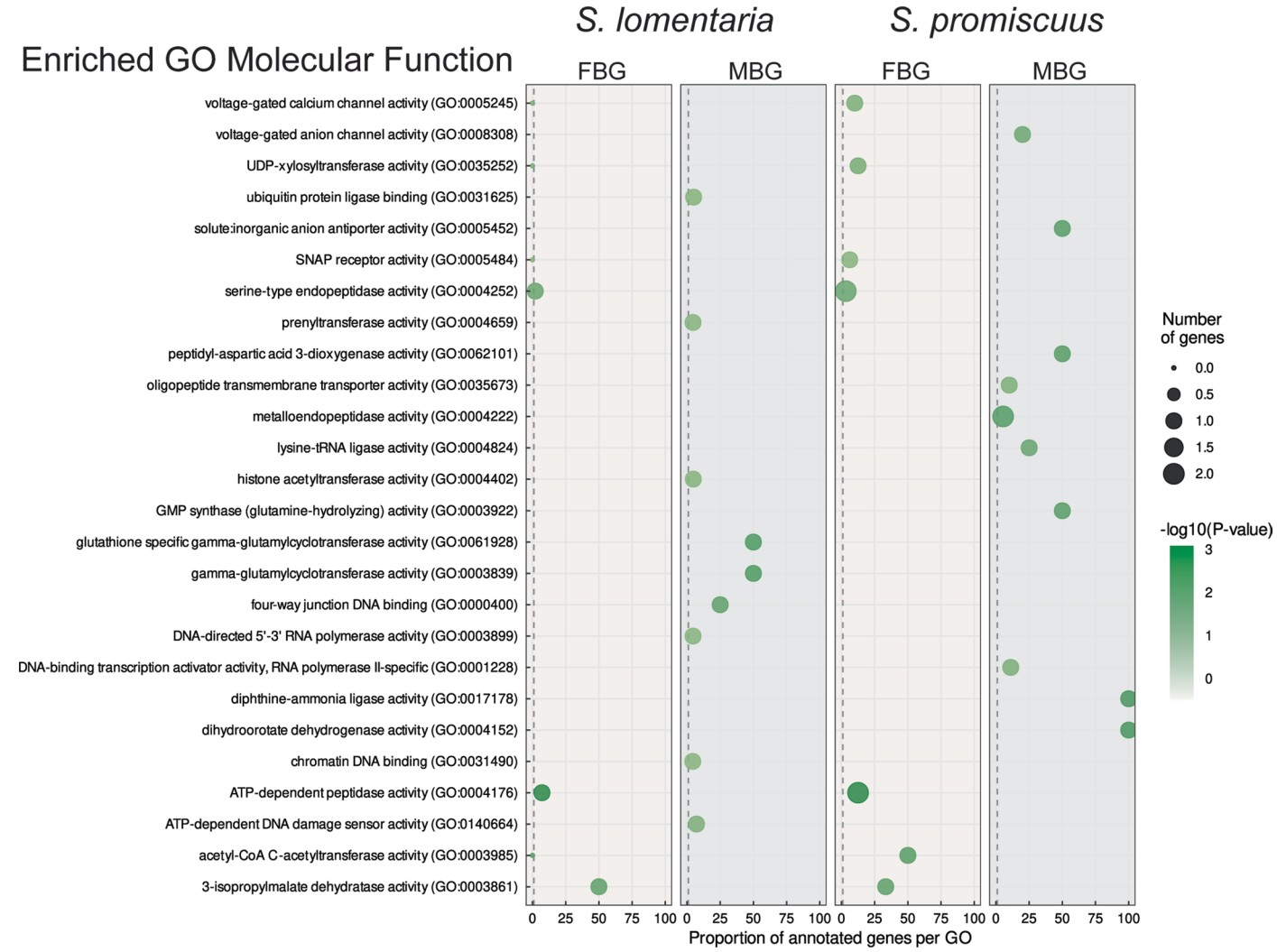

**Extended Data Fig. 2 | GO term enrichment analysis.** GO term enrichment (Molecular function) in male- and female-biased genes in *S. lomentaria* and *S. promiscuus*.

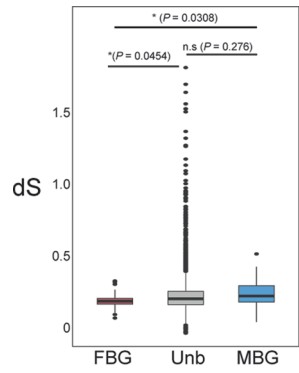
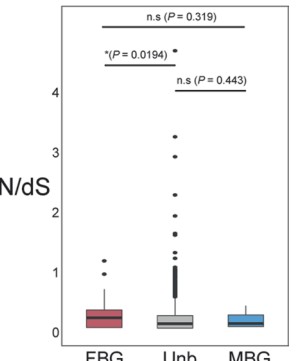

**Extended Data Fig. 3 | Evolutionary rates measured as dN/dS between species pairs (*S. lomentaria*/*S.promiscuus*) for unbiased, female-biased (FBG), and male-biased genes (MBG).** The statistical tests are permutation Mann-Whitney ranked tests two-sided; the *p*-values are displayed in parentheses. The boxes represent the interquartile ranges (25th and 75th percentiles) of the data, the lines inside the boxes represent the medians, and the whiskers represent the largest/smallest values within 1.5 times the interquartile range above and below the 75th and 25th percentiles, respectively.

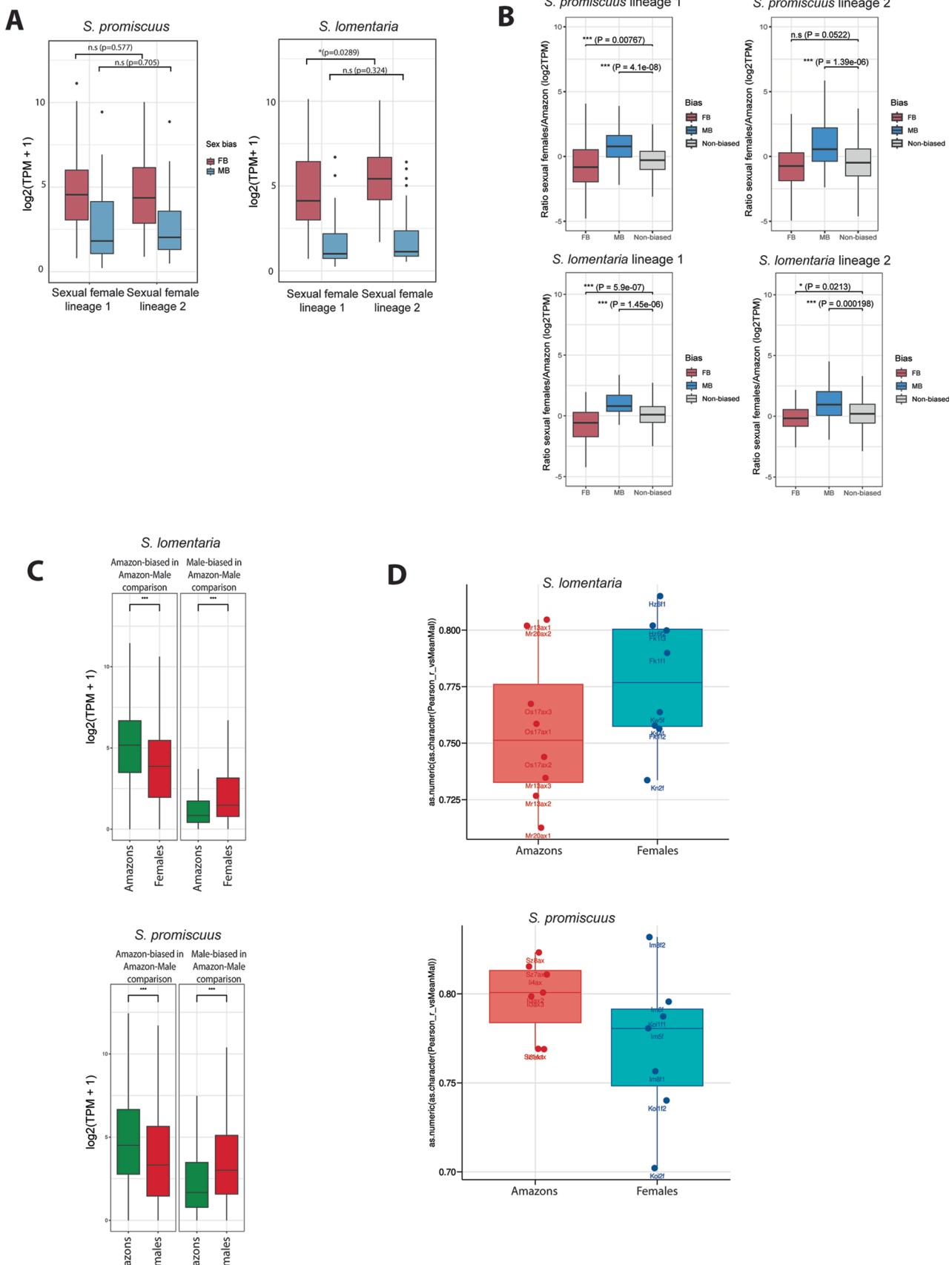

**Extended Data Fig. 4 | See next page for caption.**

**Extended Data Fig. 4 | Sex biased gene expression. A**) Changes in sex-biased gene expression when two sexual lineages are compared in *S. promiscuus* and *S. lomentaria* species. **B**) Relative gene expression comparisons between sexual females versus Amazons in each of the lineages within *S. promiscuus* and *S. lomentaria* species. The statistical tests are permutation Mann-Whitney ranked tests two-sided; the ***p***-values are displayed in parentheses. The boxes represent the interquartile ranges (25th and 75th percentiles) of the data, the lines inside the boxes represent the medians, and the whiskers represent the largest/smallest values within 1.5 times the interquartile range above and below the 75th and 25th percentiles, respectively. **C**) Expression levels of male-biased and amazon-biased genes (obtained from a comparison of gene expression between males and Amazons) in Amazons and females. **D**) Similarity in gene expression (Pearson correlations) in Amazons vs males and females vs males.

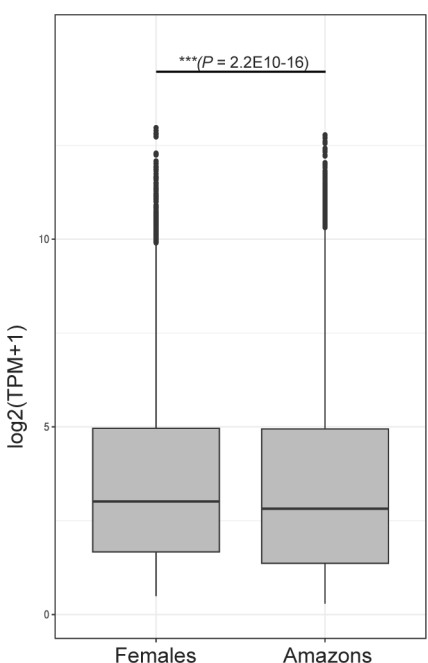

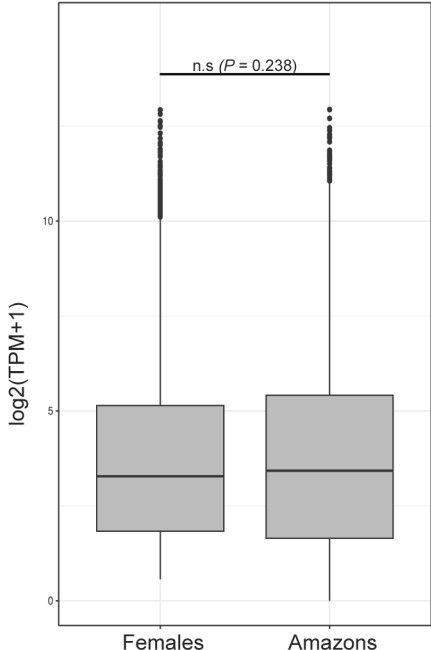

**Extended Data Fig. 5 | Comparison of unbiased gene expression levels in sexual and asexual populations, in log2(TPM + 1).** Boxes represent the interquartile range (25th and 75th percentiles) of the data, the line inside the box represents the median, whiskers represent the largest/smallest value within

1.5 times interquartile range above and below the 75th and 25th percentile, respectively. Statistical tests are Mann-Whitney ranked tests. Number of analysed genes are presented inside brackets.

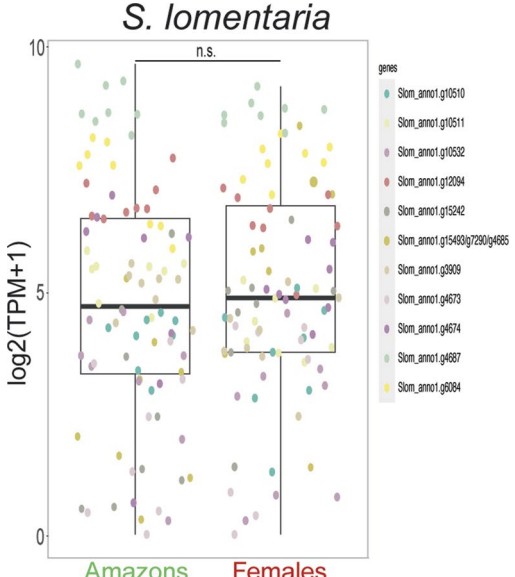

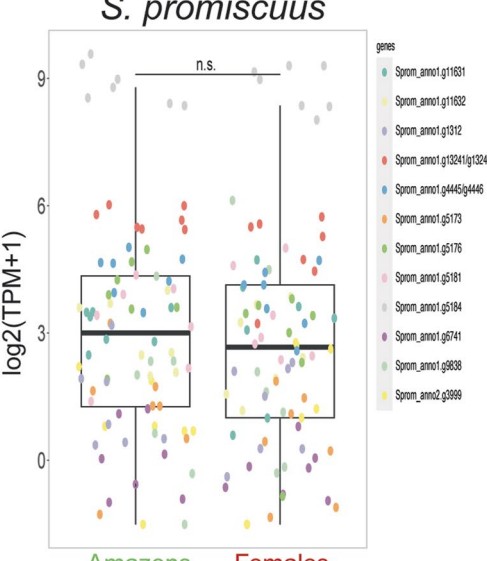

**Extended Data Fig. 6 | Mean expression level of U-linked genes in Amazon populations compared to ancestral sexual populations.** The boxes represent the interquartile ranges (25th and 75th percentiles) of the data, the lines inside the boxes represent the medians, and the whiskers represent the largest/smallest values within 1.5 times the interquartile range above and below the 75th and 25th percentiles, respectively.

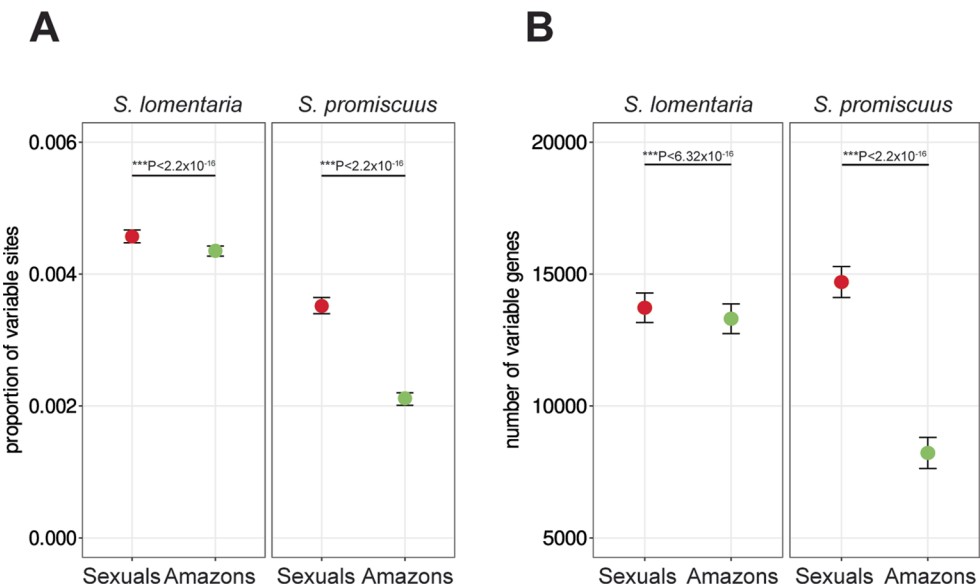

**Extended Data Fig. 7 | Selection efficacy in sexual versus Amazon populations of *S. lomentaria* and *S. promiscuus* populations.** **A**) Proportion of variable sites in sexual and Amazon populations. **B**) Number of genes presenting variable sites. The points represent the mean and the lines standard deviations. Statistical tests are Mann-Whitney ranked tests, the p-value is indicated in brackets.

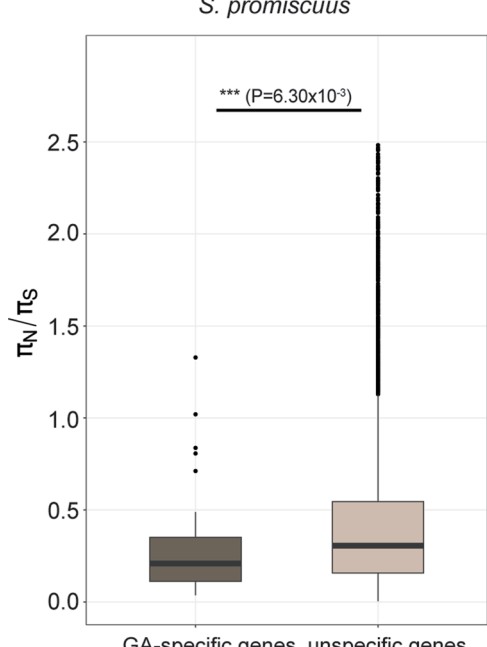

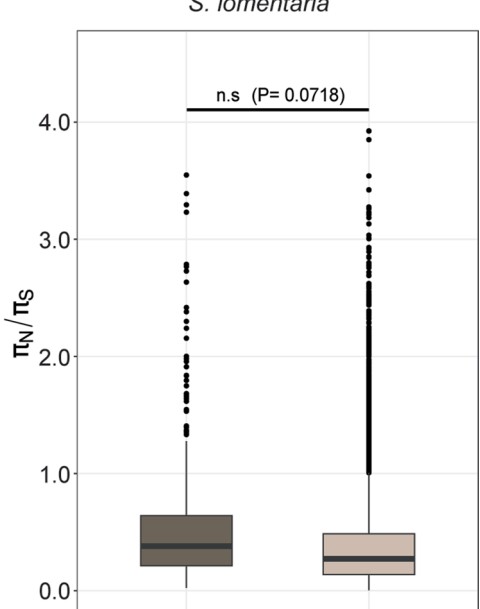

**Extended Data Fig. 8 | πN/πS in sexual populations in *S. promiscuus* and *S. lomentaria* separately for haploid (gametophyte GA)-specific genes and non-specific genes.** The boxes represent the interquartile ranges (25th and 75th percentiles) of the data, the lines inside the boxes represent the medians, and the whiskers represent the largest/smallest values within 1.5 times the interquartile range above and below the 75th and 25th percentiles, respectively.

# Reporting Summary

## Statistics

For all statistical analyses, confirm that the following items are present in the figure legend, table legend, main text, or Methods section.

| n/a | Confirmed | |
|---|---|---|
| ☐ | ☒ | The exact sample size (*n*) for each experimental group/condition, given as a discrete number and unit of measurement |
| ☒ | ☐ | A statement on whether measurements were taken from distinct samples or whether the same sample was measured repeatedly |
| ☐ | ☒ | The statistical test(s) used AND whether they are one- or two-sided<br>*Only common tests should be described solely by name; describe more complex techniques in the Methods section.* |
| ☐ | ☒ | A description of all covariates tested |
| ☐ | ☒ | A description of any assumptions or corrections, such as tests of normality and adjustment for multiple comparisons |
| ☐ | ☒ | A full description of the statistical parameters including central tendency (e.g. means) or other basic estimates (e.g. regression coefficient) AND variation (e.g. standard deviation) or associated estimates of uncertainty (e.g. confidence intervals) |
| ☐ | ☒ | For null hypothesis testing, the test statistic (e.g. $F$, $t$, $r$) with confidence intervals, effect sizes, degrees of freedom and $P$ value noted<br>*Give P values as exact values whenever suitable.* |
| ☒ | ☐ | For Bayesian analysis, information on the choice of priors and Markov chain Monte Carlo settings |
| ☐ | ☒ | For hierarchical and complex designs, identification of the appropriate level for tests and full reporting of outcomes |
| ☐ | ☒ | Estimates of effect sizes (e.g. Cohen's *d*, Pearson's *r*), indicating how they were calculated |

*Our web collection on statistics for biologists contains articles on many of the points above.*

## Software and code

Policy information about availability of computer code

| Data collection | na |
|---|---|
| Data analysis | All software incl. version number used in this study are mentioned at the manuscript. No custom code was used. |

For manuscripts utilizing custom algorithms or software that are central to the research but not yet described in published literature, software must be made available to editors and reviewers. We strongly encourage code deposition in a community repository (e.g. GitHub). See the Nature Portfolio guidelines for submitting code & software for further information.

## Data

Policy information about availability of data

All manuscripts must include a data availability statement. This statement should provide the following information, where applicable:

- Accession codes, unique identifiers, or web links for publicly available datasets
- A description of any restrictions on data availability
- For clinical datasets or third party data, please ensure that the statement adheres to our policy

All data has been deposited in data repositories and accession codes are given in the manuscript

# Research involving human participants, their data, or biological material

Policy information about studies with [human participants or human data](). See also policy information about [sex, gender (identity/presentation), and sexual orientation]() and [race, ethnicity and racism]().

| | |
|---|---|
| Reporting on sex and gender | na |
| Reporting on race, ethnicity, or other socially relevant groupings | na |
| Population characteristics | na |
| Recruitment | na |
| Ethics oversight | na |

Note that full information on the approval of the study protocol must also be provided in the manuscript.

# Field-specific reporting

Please select the one below that is the best fit for your research. If you are not sure, read the appropriate sections before making your selection.

☐ Life sciences  ☐ Behavioural & social sciences  ☒ Ecological, evolutionary & environmental sciences

For a reference copy of the document with all sections, see [nature.com/documents/nr-reporting-summary-flat.pdf]()

# Ecological, evolutionary & environmental sciences study design

All studies must disclose on these points even when the disclosure is negative.

| | |
|---|---|
| Study description | all details are provided in the manuscript |
| Research sample | all samples are described in the material and methods and supplemental tables |
| Sampling strategy | samples were collected from the field using standard approaches described in methods section. |
| Data collection | M. Hoshino recorded data, all details are provided in Supplemental Data |
| Timing and spatial scale | all information is available in Suppl. tables |
| Data exclusions | no data was excluded |
| Reproducibility | all attempts were successfull |
| Randomization | petri dishes containing the samples were randomly distributed in the culture chambers until they were collected for RNA extraction |
| Blinding | samples were collected from the field without previous knowledge of their sex. Pheromone tests were performed blindly. |

Did the study involve field work?  ☒ Yes  ☐ No

## Field work, collection and transport

| | |
|---|---|
| Field conditions | Sampling were conducted at sea coasts listed in Table S1, in spring |
| Location | shown in Figure 1B and Table S1. |
| Access & import/export | All samples were collected in Japan and were imported to Germany following Japanese and German laws and Nagoya protocol. |
| Disturbance | non destructive sampling |

# Reporting for specific materials, systems and methods

We require information from authors about some types of materials, experimental systems and methods used in many studies. Here, indicate whether each material, system or method listed is relevant to your study. If you are not sure if a list item applies to your research, read the appropriate section before selecting a response.

## Materials & experimental systems

| n/a | Involved in the study |
|-----|----------------------|
| ☒ ☐ | Antibodies |
| ☒ ☐ | Eukaryotic cell lines |
| ☒ ☐ | Palaeontology and archaeology |
| ☒ ☐ | Animals and other organisms |
| ☒ ☐ | Clinical data |
| ☒ ☐ | Dual use research of concern |
| ☒ ☐ | Plants |

## Methods

| n/a | Involved in the study |
|-----|----------------------|
| ☒ ☐ | ChIP-seq |
| ☒ ☐ | Flow cytometry |
| ☒ ☐ | MRI-based neuroimaging |

## Plants

| | |
|---|---|
| Seed stocks | na |
| Novel plant genotypes | na |
| Authentication | na |

