## [Peer Review File · Nature Ecology & Evolution]

PARALLEL LOSS OF SEXUAL REPRODUCTION IN FIELD POPULATIONS OF A BROWN ALGA SHEDS LIGHT ON THE MECHANISMS UNDERLYING THE EMERGENCE OF ASEXUALITY

Corresponding Author: Dr Susana Coelho

Version 0:

Decision Letter:

31st January 2024

Dear Susana,,

Thank you very much for your enquiry about submitting your manuscript "PARALLEL LOSS OF SEX IN FIELD POPULATIONS OF A BROWN ALGA SHEDS LIGHT ON THE MECHANISMS UNDERLYING THE EMERGENCE OF ASEXUALITY" to Nature Ecology & Evolution. It certainly sounds interesting, and we would be happy to consider it for publication. We were a bit unsure about implications of these findings outside the Amazon system but on balance we thought that a first characterisation of evolution of assexuality in a haploid system is interesting to check with reviewers.

In order to submit your complete manuscript to Nature Ecology & Evolution, please use the link below:

Link Redacted

If you have any questions, please feel free to contact me.

[REDACTED]

Version 1:

Decision Letter:

12th March 2024

Dear Susana,

Your manuscript entitled "PARALLEL LOSS OF SEX IN FIELD POPULATIONS OF A BROWN ALGA SHEDS LIGHT ON THE MECHANISMS UNDERLYING THE EMERGENCE OF ASEXUALITY" has now been seen by three reviewers, whose comments are attached. The reviewers have raised a number of concerns which will need to be addressed before we can offer publication in Nature Ecology & Evolution. We will therefore need to see your responses to the criticisms raised and to some editorial concerns, along with a revised manuscript, before we can reach a final decision regarding publication.

We encourage you to rewrite the Discussion following Reviewer #1's recommendations to put your results in a broader context and discuss implications for the evolution of assexuality in general.

Please make sure the Abstract is no longer than 200 words.

We therefore invite you to revise your manuscript taking into account all reviewer and editor comments. Please highlight all changes in the manuscript text file in Microsoft Word format.

* If you have not done so already please begin to revise your manuscript so that it conforms to our Article format instructions at <http://www.nature.com/natecolevol/info/final-submission>. Refer also to any guidelines provided in this letter.

Link Redacted

Nature Ecology & Evolution is committed to improving transparency in authorship. As part of our efforts in this direction, we are now requesting that all authors identified as 'corresponding author' on published papers create and link their Open Researcher and Contributor Identifier (ORCID) with their account on the Manuscript Tracking System (MTS), prior to acceptance. ORCID helps the scientific community achieve unambiguous attribution of all scholarly contributions. You can create and link your ORCID from the home page of the MTS by clicking on 'Modify my Springer Nature account'. For more information please visit please visit www.springernature.com/orcid.

[REDACTED]

Reviewer expertise:

Reviewer #1: evolution of asexuality, transcriptomics

Reviewer #2: evolution of asexuality, transcriptomics

Reviewer #3: evolution of reproductive strategies, including parthenogenesis

Reviewers' comments:

Reviewer #1 (Remarks to the Author):

This study investigates traits and gene expression patterns associated with transitions to obligate asexuality in brown algae, in which males and females are haploid and multicellular. Asexual reproduction occurs through the development of unfertilized female gametes. Obligate asexuality is accompanied by a loss of the production of a pheromone by which female gametes attract male gametes in sexual populations, as well as by traits that likely improve the parthenogenetic development (i.e., development without fertilization) of these gametes. Obligate asexuality is also accompanied by a "masculinization" of gene expression, that is, by an average change in the expression of sex-biased genes in the direction of the male expression profile, as has been observed also in some asexual animals. Finally, the study identifies a candidate autosomal gene that may be directly involved in the transition, though its precise function remains unclear (and distinguishing cause from effect is difficult, as also for the other traits associated with asexuality).

The study has three main strengths. First, it approaches transitions to obligate asexuality from multiple angles, including reproductive phenotypes, gene expression, and possible underlying genetic mechanisms. Second, it demonstrates that the above associations occur in a repeated fashion in three, at least partly independent transitions to obligate asexuality (two transitions within one species and a parallel transition in a related species). Third, studying asexuality in brown algae adds an important component of generality to existing studies on these transitions, as most other studies were carried out on phylogenetically distant plants and animals.

These strengths, together with the broad interest of the topic (evolution of asexuality) within Ecology and Evolution (and beyond), makes this manuscript a highly suitable candidate for publication in a journal with a broad and general reach. In addition, the study is data-rich, the findings well supported, and the manuscript generally well-written. Yet, I feel that a few points may need clarification and a few sections may need to be revised to gain even more generality and thus an even further reach.

First, the discussion mostly focuses on findings specific to this particular study system and does relatively little to explain the generality of the results and to capitalize on the phylogenetic outlier situation of this study (an exception is the discussion of the masculinization of gene expression). What else can be learnt on the evolution of asexuality and its mechanistic underpinnings, in general? To me, some of the results are reminiscent of aphids, which evolved obligate asexuality by losing the ability to respond to environmental stimuli initiating sex. In both cases, obligate asexuality evolves from already existing pathways, a situation that may be rather different from evolving an entirely novel asexual pathway, in which life cycle features that usually are connected to meiosis (e.g., epigenetic resetting) need to be readjusted. This is of relevance here, as it does not become clear how supposedly 1n partheno-sporophytes (see also specific comment on Fig. 1 below) produce spores. Is "apo-meiosis" a mitosis or is it a meiosis with first-division restitution (as in males of haplodiploid insects) or normal meiosis preceded by an additional doubling step (as in hemi-clonal frogs)? These details may be unimportant, but it is unclear how spore production occurs without any additional evolutionary change unless a spore-production pathway specific to partheno-sporophytes was already present at the time of the transition to obligate asexuality. This means that without an asexuality pathway already present, loss of pheromone production of female gametes (plus traits improving parthenogenetic development of these gametes) would likely be insufficient for the evolution of an entirely asexual life cycle.

Second, the discussion of sexual conflict over gene expression is somewhat confusing. It seems like an additional way to set up the paper (in a similar way to other papers in the literature), saying that studying gene expression in female-only populations allows testing whether there is a conflict over gene expression between males and females. Indeed, the finding of masculinization of gene expression in absence of males suggests that sexual conflict does not lead reduced (i.e., less than optimal expression differences between males and females. However, it is also possible that sexual conflict leads to an exaggeration of expression differences, e.g., because females may overexpress or down-regulate certain genes to combat genetic conflict from males. Furthermore, there may be genetic conflict about the expression of genes that are, for some reason, not regulated differently (i.e., there is no expression bias). Hence, the conclusion that there is no sexual conflict over gene expression does not seem warranted and this discussion does not seem to add much to the paper (perhaps it can still be pointed out, as similar arguments are found in the literature). The discussion may also be more directly linked to the observed changes in phenotypes, some of which are more male-like in Amazons (e.g., absence of pheromone), others more strongly pronounced female (e.g., larger gamete size). The results indicate that, for gene expression, the former category outweighs the latter, perhaps linked to degeneration of costly traits mediating male-female interactions in sexual populations.

Specific comments:

- Connected to point 1 above, the question of independent evolution is perhaps glanced over a bit quickly. Is there really no gene-flow back to sexual populations even in the contact zones? And have these asexual populations remained stable in place since the split from the sexuals? Even rare and limited gene flow may lead to introgression of genes linked to asexuality and to the re-use of the same genes in repeated transitions to asexuality, blurring the concept of independence in these events of parallel evolution.
- Fig. 1: Give ploidies also for partheno-sporophyte, explain apomeiosis, and say why prefixes "apo" and "partheno" are sometimes in brackets.
- L. 126 typo "rex"
- L. 347-349: Why not just give $P_i(s)$?
- L. 365 typo: one of the instances of Amazons should be sexual populations.
- L. 377: One might expect to see the strongest effects in sporophyte-specific genes, as these are 2n in sexual populations and 1n in Amazons. Are there any such genes?
- Difference in selection patterns between species: Many factors may contribute: difference in number of sporophyte-specific genes, time since transition to asexuality (given large confidence intervals on time estimates), different degrees of geneflow between sexual populations and Amazons, etc.

Reviewer #2 (Remarks to the Author):

In this study, Hoshino et al. study two species of brown alga *Scytosiphon* to investigate the consequences of the evolution of asexuality from a sexual ancestor that occurred independently in those two species. The authors find that, in line with other studies on the loss of sex, the transcriptome of asexual females is defeminized as well as masculinized. This indicates that not sexual antagonism but decay of female functions drives gene expression patterns in asexuals. In agreement with this, the pheromone production in females has been lost convergently in asexuals of both of these species. In this study, the authors investigate this further by showing that asexual gametes are larger and develop faster while still retaining the ability to recognise and fuse with male gametes, except for one population where the transition to asexuality seems to be further developed. In contrast to data from stick insects and theoretical predictions, relaxed purifying selection could not be found in these algae and the authors hypothesize that this may be due to the haplo-diplontic lifestyle and purifying selection being more efficient in the haploid life stage to counteract mutational meltdown.

I really like this study. The analyses are well thought through and very well analysed. The paper is written in a very streamlined way and the argumentation and reasoning for the different analyses & methods is easy to follow. The effect of selection at haploid life stages is a special feature of this study system and has not been investigated well enough yet. Hence, this work increases our understanding of the evolution of reproductive modes and its causes and consequences on the molecular as well as on the genetic level. This study is definitely interesting and relevant for a wider audience in evolutionary biology. I recommend accepting this paper with only some very minor comments that I trust the authors to address:

- Fig. 1: Colour codes in C) I do not understand. What is meant with gamete behaviour phenotypes scored in males/females/asexuals? I understand that the green boxes show the obligate asexual populations but what distinguishes red and blue is not quite clear to me.
- Fig. 2: B-E it would be nice to have the 2 sub-clades named and then these names above the 2 different panels as it is very hard to keep the lineage abbreviations in mind and in the beginning, I thought the two panels were about the 2 species rather than the 2 *S. promiscuus* clades.
- Fig.2 C): I do not understand the number of green/red dots. I thought the numbers in italics represent the number of scored lineages but if that is the case, then why does lineage li6ax have 5 green dots at zero and lineage li9ax just 1?
- L. 282: This belongs to A) rather than to B).
- SBGs based on all sexual populations? What happens if called separately in each population? Does the pattern hold or are then more SBGs detectable? Does the choice of reference influence the conclusions about de-feminisation and masculinisation?
- Fig. S4: Is this not a bit of a circular argument? You anyway use these populations to define SBGs and then find that FBGs are higher expressed in females than in males. To really test whether this is an artifact of SBG turnover, it would be more meaningful to compare the sexual lineages of the two species against one another using first the SBG classification based on *S. promiscuus* and then that based on *S. lomentaria*.

Reviewer #3 (Remarks to the Author):

This paper looks at phenotypic and genetic changes following the loss of sex in a brown algae. The paper addresses a key question in evolutionary biology, and is well written and analysed. I have only a few comments.

Firstly, it would be good to provide more information on the pheromone production. At line 441 you say it is complex, but up until then I thought it could be relatively simple? I.e. caused by a null mutation or something? You could look in your data for this, either by looking for genes with null mutations in the amazons, or genes with zero expression in amazons but with high expression in females (I am not sure if you filtered these out?)

The authors have produced a lot of genetic resources (Genome assemblies, annotation, RNAseq), but their data does not seem to be available? In addition, they have performed complex analyses, but not provided their code. This should be provided either as supplementary material or in a digital repository (e.g. Zotero).

Minor

Line 57 – I think that you should refer to Parker, D. J., Bast, J., Jalvingh, K., Dumas, Z., Robinson-Rechavi, M., Schwander, T. 2019. Repeated evolution of asexuality involves convergent gene expression changes. *Molecular Biology and Evolution*. 36: 350–364 rather than Parker, D. J., Bast, J., Jalvingh, K., Dumas, Z., Robinson-Rechavi, M., Schwander, T. 2019. Sex-biased gene expression is repeatedly masculinized in asexual females. *Nature Communications*. 10: 4638 here.

Line 126 – ratio not ration

Fig 1C – please make it clear which lineages are being compared here.

Line 195 – ‘fully asexual’ – do you mean ‘obligately asexual’?

Line 696 – ‘gens’ should be genes.

*****END*****

Author Rebuttal letter:

Reviewers' comments:

Reviewer #1 (Remarks to the Author):

This study investigates traits and gene expression patterns associated with transitions to obligate asexuality in brown algae, in which males and females are haploid and multicellular. Asexual reproduction occurs through the development of unfertilized female gametes. Obligate asexuality is accompanied by a loss of the production of a pheromone by which female gametes attract male gametes in sexual populations, as well as by traits that likely improve the parthenogenetic development (i.e., development without fertilization) of these gametes. Obligate asexuality is also accompanied by a masculinization of gene expression, that is, by an average change in the expression of sex-biased genes in the direction of the male expression profile, as has been observed also in some asexual animals. Finally, the study identifies a candidate autosomal gene that may be directly involved in the transition, though its precise

function remains unclear (and distinguishing cause from effect is difficult, as also for the other traits associated with asexuality).

The study has three main strengths. First, it approaches transitions to obligate asexuality from multiple angles, including reproductive phenotypes, gene expression, and possible underlying genetic mechanisms. Second, it demonstrates that the above associations occur in a repeated fashion in three, at least partly independent transitions to obligate asexuality (two transitions within one species and a parallel transition in a related species). Third, studying asexuality in brown algae adds an important component of generality to existing studies on these transitions, as most other studies were carried out on phylogenetically distant plants and animals.

These strengths, together with the broad interest of the topic (evolution of asexuality) within Ecology and Evolution (and beyond), makes this manuscript a highly suitable candidate for publication in a journal with a broad and general reach. In addition, the study is data-rich, the findings well supported, and the manuscript generally well-written. Yet, I feel that a few points may need clarification and a few sections may need to be revised to gain even more generality and thus an even further reach.

First, the discussion mostly focuses on findings specific to this particular study system and does relatively little to explain the generality of the results and to capitalize on the phylogenetic outlier situation of this study (an exception is the discussion of the masculinization of gene expression). What else can be learnt on the evolution of asexuality and its mechanistic underpinnings, in general? To me, some of the results are reminiscent of aphids, which evolved obligate asexuality by losing the ability to respond to environmental stimuli initiating sex. In both cases, obligate asexuality evolves from already existing pathways, a situation that may be rather different from evolving an entirely novel asexual pathway, in which life cycle features that usually are connected to meiosis (e.g., epigenetic resetting) need to be readjusted. This is of relevance here, as it does not become clear how supposedly 1n partheno-sporophytes (see also specific comment on Fig. 1 below) produce spores. Is apomeiosis a mitosis or is it a meiosis with first-division restitution (as in males of haplodiploid insects) or normal meiosis preceded by an additional doubling step (as in hemi-clonal frogs)? These details may be unimportant, but it is unclear how spore production occurs without any additional evolutionary change unless a spore-production pathway specific to partheno-sporophytes was already present at the time of the transition to obligate asexuality. This means that without an asexuality pathway already present, loss of pheromone production of female gametes (plus traits improving parthenogenetic development of these gametes) would likely be insufficient for the evolution of an entirely asexual life cycle.

Re: We thank the reviewer for the overall positive comments and for the excellent suggestions. We have revised the discussion according to the recommendations, and added more considerations about emergence of obligate parthenogenesis from a pre-existing trait (facultative parthenogenesis), i.e., the fact that asexuality in both animals and brown algae builds on a trait that is already there, so probably easier to evolve.

Concerning the reviewer's comment on whether 1n individuals go through meiosis (i.e. how they produce meiospores): we currently have little knowledge on the mechanisms underlying this so-called apomeiosis. We have previously shown that parthenosporophytes of a closely related species (*Ectocarpus*) are mostly haploid, although some cells may endoreduplicate during development. We suspect that the cell that differentiates into the unilocular sporangia (the structure where spores are produced via meiosis) may undergo endoreduplication, becomes 2n and then a classical meiosis may occur. Alternatively, a non-reductive meiosis could occur if the unilocular sporangia remains haploid. It is currently very difficult to distinguish between these options, and more generally, to study apomeiosis in these organisms, because imaging is difficult and transformation with GFP or tagged proteins is not possible. We believe that a genome reprogramming occurs in the unilocular sporangia, and this reprogramming is required for the meiospores to engage in the gametophyte program. We therefore use the general term apomeiosis to describe modified meiosis that occurs in the unilocular sporangia and that allows the switch to the gametophyte generation.

Regardless of the means to produce the spores, we agree with the reviewer and we confirm that the spore-production pathway specific to partheno-sporophytes is already present at the time of the transition to obligate asexuality. We added a sentence to explain these ideas more thoroughly in the new manuscript:

In parallel with this sexual life cycle, brown algae may have an asexual life cycle in which unfused female (and occasionally male) gametes undergo parthenogenesis to develop into adult multicellular individuals (Peters et al., 2008; Bothwell et al., 2010a,b; Arun et al., 2013; Mignerot & Coelho, 2016; Mignerot et al., 2019). The events underlying haploid spore production from (presumably haploid) parthenosporophytes is currently elusive, but are thought to involve apomeiosis (either a non-reductive meiosis or endoreduplication followed by meiosis) (Bothwell et al 2010).

Second, the discussion of sexual conflict over gene expression is somewhat confusing. It seems like an additional way to set up the paper (in a similar way to other papers in the literature), saying that studying gene expression in female-only populations allows testing whether there is a conflict over gene expression between males and females. Indeed, the finding of masculinization of gene expression in absence of males suggests that sexual conflict does not lead to reduced (i.e., less than optimal expression differences between males and females). However, it is also possible that sexual conflict leads to an exaggeration of expression differences, e.g., because females may overexpress or down-regulate certain

genes to combat genetic conflict from males. Furthermore, there may be genetic conflict about the expression of genes that are, for some reason, not regulated differently (i.e., there is no expression bias). Hence, the conclusion that there is no sexual conflict over gene expression does not seem warranted and this discussion does not seem to add much to the paper (perhaps it can still be pointed out, as similar arguments are found in the literature). The discussion may also be more directly linked to the observed changes in phenotypes, some of which are more male-like in Amazons (e.g., absence of pheromone), others more strongly pronounced female (e.g., larger gamete size). The results indicate that, for gene expression, the former category outweighs the latter, perhaps linked to degeneration of costly traits mediating male-female interactions in sexual populations.

Re: We thank the reviewer to pointing out these ideas. The reviewer is correct: we discussed sexual conflict in our manuscript indeed because similar arguments have been used in the extensive (animal) literature, and we thought it was important to mention this aspect of our results, although we agree this does not add too much to whole story. So we would prefer to keep those considerations in the manuscript. Concerning linking the observed changes in phenotype with gene expression, this is an excellent point: we have followed the reviewer's advice and added a section in the discussion linking the changed phenotypes, the gene expression patterns and the degeneration of costly traits mediating male-female interactions in sexual populations:

Finally, we also note the complexity of the Amazon phenotype, in that some traits are more male-like (absence of pheromone), corresponding to masculinization, whereas others are more pronounced female-traits (large gamete size), i.e., corresponding to feminization. Our results indicate that, for gene expression, the former category outweighs the latter, perhaps reflecting a degeneration of costly traits mediating male-female interactions in sexual populations.

Specific comments:

- Connected to point 1 above, the question of independent evolution is perhaps glanced over a bit quickly. Is there really no gene-flow back to sexual populations even in the contact zones? And have these asexual populations remained stable in place since the split from the sexuals? Even rare and limited gene flow may lead to introgression of genes linked to asexuality and to the re-use of the same genes in repeated transitions to asexuality, blurring the concept of independence in these events of parallel evolution.

Re: We found that the Amazon phenotype arose in the two distinct species, which are clearly divergent and these species show no sign of hybridization. Therefore, our data are clearly consistent with least two independent origins of asexuality. Note that complete pre-zygotic barrier has been confirmed between *S. lomentaria* and *S. promiscuus* (Hoshino et al. 2018: <https://doi.org/10.2216/17-77.1>). So, no gene flow is expected between the two species, although they may have overlapping geographic distributions.

For the particular case of *S. promiscuus* Amazon li and sexual population lm, they are parapatric (1 km apart), but our (unpublished) genome wide SNPs analyses did not detect any gene flow between them (most likely because li females have lost fertilization capacity, see Figure 1C). Therefore, we conclude that gene flow would be extremely unlikely between *S. promiscuus* sexual and Amazon populations.

Nevertheless, we have been very conservative by writing that two origins of asexuality in this species are likely and in the new version of the manuscript, we further state that we cannot completely rule out that asexuality could have been introduced through hybridization:

" RNA-seq data was used to build a phylogenetic tree, which showed a clear separation of the *S. lomentaria* and *S. promiscuus* lineages, and revealed that Amazons emerged from their sexual ancestors repeatedly and independently in each species, with likely two independent origins of Amazon lineages within *S. promiscuus* (Figure 1C), even if we cannot fully exclude that genes involved in asexuality could have been introduced from one lineage to the other via introgression."

Importantly, our analysis (DILS, see suppl. data) only detected potential gene flow between sexual and Amazons within each three lineages (within *S. lomentaria*, within *S. promiscuus* li, and within *S. promiscuus* lm). Therefore, even if it is possible, we didn't propose that Amazons originated several times within these three lineages. That's why we stated that the results support at least two or three independent origins.

- Fig. 1: Give ploidies also for partheno-sporophyte, explain apomeiosis, and say why prefixes apo and partheno are sometimes in brackets.

- L. 126 typo rex

Re: This has been done/corrected

- L. 347-349: Why not just give Pi(s)?

Re: We have added the following sentence in the text: Accordingly there was a significant difference in Pis between sexual and asexual populations in both species (Fig 5B).

- L. 365 typo: one of the instances of Amazons should be sexual populations.

Re: the typo has been corrected.

- L. 377: One might expect to see the strongest effects in sporophyte-specific genes, as these are 2n in sexual populations and 1n in Amazons. Are there any such genes?

Re: There are indeed such genes in the life cycle, i.e., genes that are expressed only during sporophyte stage (sporophyte-specific genes). We identified 87 sporophyte-specific genes in *S. lomentaria* and 209 sporophyte-specific genes in *S. promiscuus*. However, our idea was to compare genes under haploid selection whenever they are expressed (i.e., gametophyte-specific genes) with genes that are otherwise alternating between diploid and haploid life stages, (i.e., the great majority of genes in the sexual *Scytosiphon*). Comparing the substitution rates between gametophyte-specific and sporophyte-specific genes (as the reviewer suggests) would indeed certainly show an even stronger difference, exacerbating the effect, but that would not inform us about the differences observed between the asexuals (always haploid) and sexual (alternating haploid and diploid life stages). So we would prefer to use our more conservative approach, that still shows a clear role for haploid selection.

- Difference in selection patterns between species: Many factors may contribute: difference in number of sporophyte-specific genes, time since transition to asexuality (given large confidence intervals on time estimates), different degrees of geneflow between sexual populations and Amazons, etc.

Re: We agree with the reviewer that many factors can explain the difference in selection between the two species. We have now rephrased this aspect more clearly:

âRemarkably, our study suggests that selection is not less efficient in Amazon populations compared with the sexual counterparts, and this may be due to haploid purifying selection overwhelming the relaxation of selection due to asexuality, and avoiding the deleterious consequences of prolonged asexuality (Neiman et al., 2010; Bast et al., 2018; HÃ¶randl et al., 2020). Note that additional factors such as different age of asexuality or different magnitude of gene flow between sexual and asexuals can also contribute to the different patterns observed between the two species.â

Reviewer #2 (Remarks to the Author):

In this study, Hoshino et al. study two species of brown alga *Scytosiphon* to investigate the consequences of the evolution of asexuality from a sexual ancestor that occurred independently in those two species. The authors find that, in line with other studies on the loss of sex, the transcriptome of asexual females is defeminized as well as masculinized. This indicates that not sexual antagonism but decay of female functions drives gene expression patterns in asexuals. In agreement with this, the pheromone production in females has been lost convergently in asexuals of both of these species. In this study, the authors investigate this further by showing that asexual gametes are larger and develop faster while still retaining the ability to recognise and fuse with male gametes, except for one population where the transition to asexuality seems to be further developed. In contrast to data from stick insects and theoretical predictions, relaxed purifying selection could not be found in these algae and the authors hypothesize that this may be due to the haplo-diplontic lifestyle and purifying selection being more efficient in the haploid life stage to counteract mutational meltdown.

I really like this study. The analyses are well thought through and very well analysed. The paper is written in a very streamlined way and the argumentation and reasoning for the different analyses & methods is easy to follow. The effect of selection at haploid life stages is a special feature of this study system and has not been investigated well enough yet. Hence, this work increases our understanding of the evolution of reproductive modes and its causes and consequences on the molecular as well as on the genetic level. This study is definitely interesting and relevant for a wider audience in evolutionary biology. I recommend accepting this paper with only some very minor comments that I trust the authors to address:

â Fig. 1: Colour codes in C) I do not understand. What is meant with gamete behaviour phenotypes scored in males/females/asexuals? I understand that the green boxes show the obligate asexual populations but what distinguishes red and blue is not quite clear to me.

Re: We have added more details in the legend on the figure. Red represent females from sexual populations, blue males from sexual populations.

â Fig. 2: B-E it would be nice to have the 2 sub-clades named and then these names above the 2 different panels as it is very hard to keep the lineage abbreviations in mind and in the beginning, I thought the two panels were about the 2 species rather than the 2 *S. promiscuus* clades.

Re: this has been done.

â Fig.2 C): I do not understand the number of green/red dots. I thought the numbers in italics represent the number of scored lineages but if that is the case, then why does lineage li6ax have 5 green dots at zero and lineage li9ax just 1?

Re: The numbers inside brackets represent the number of scored gametes (not the number of scored lineages) in each lineage. Each point in the graph represents the fertilisation success of one independent cross. So five green dots at zero mean 5 different crosses where no zygotes were obtained. This is indicated in the legend.

â L. 282: This belongs to A) rather than to B).

Re: This has been checked.

â SBGs based on all sexual populations? What happens if called separately in each population? Does the pattern hold or are then more SBGs detectable? Does the choice of reference influence the conclusions about de-feminisation and masculinisation?

Re: We thank the reviewer for the suggestions. The aim of our analysis was to infer sex-bias gene expression levels for each of the *Scytosiphon* species, and not to focus on population-specific or lineage specific sex-differences. Our analysis of SBG was therefore performed in each species separately, using all lineages within each species. We believe that our approach of pooling together all samples from the same species is more appropriate than treating the data per lineage or population, and especially considering the comments of Reviewer 1 about the fact that we cannot fully exclude potential introgression between populations within the same species.

This approach also allows to compute a higher number of samples, reinforcing the statistical analysis. Importantly, note that the variance in the samples is in any case taken in consideration by the algorithm DEseq2 for the inference of sex-bias.

Concerning the choice of reference: the way we computed the comparison was to take the FBG and MBG in the Amazon populations and compare their expression level to the (ancestral) females, separately in each *Slom* and *Sprom* species. So, in other words, our reference was always the ancestral sexual female population. To address the reviewer comments, we now also include an analysis of Amazons versus sexuals per lineage (new Figure S4B). The choice of the reference does not change the conclusions, in other words, when we compare Amazons with sexual females per lineage within each species we still, see a de-feminisation and a masculinisation of the Amazons. This result is now included in the manuscript as Figure S4B.

Fig. S4: Is this not a bit of a circular argument? You anyway use these populations to define SBGs and then find that FBGs are higher expressed in females than in males. To really test whether this is an artifact of SBG turnover, it would be more meaningful to compare the sexual lineages of the two species against one another using first the SBG classification based on *S. promiscuus* and then that based on *S. lomentaria*.

Re: We thank the reviewer for the suggestion. As explained above, SBG in each species were defined based on analysis of all populations within each species (and not per lineage), and as Reviewer 1 highlights, we cannot fully exclude introgression within lineages of the same species, so we think it makes more sense to perform the analysis per species (and not per lineage). We did as suggested and added also the results when individual lineages are analysed separately (Figure S4B).

Moreover, note that we have a low number of SBG (which is very common in brown algae because they have low level of sexual dimorphism). Therefore, the proposed analysis across species would be very challenging to perform because we have only a small number of single copy orthologs that are SBG across the two species, limiting our power to do the proposed analysis.

Our analysis illustrated in Figure 3E shows that in both *Slom* and *Sprom* species, FBG are downregulated and MBG are upregulated in Amazons lineages compared to females, which corresponds to a defeminisation and masculinisation of gene expression in the Amazons (female -> amazon). In order to confirm that this defeminization / masculinisation of gene expression is specific to the transition to asexuality, we also compared FBG and MBG expression levels in pairs of sexual lineages (female->female) (Figure S4). Moreover, the analyses presented in Fig. S4 also allowed us to test if the trends were lineage-specific or if all lineages within each species behaved similarly. Figure S4 shows that sexual lineages (within each species) have the same level of expression of FBG and MBG indicating that the defeminisation / masculinisation trends we see are specific to the transition to sexual->amazon. Again, note that the definition of SBG was done globally per species (and not per population) so we believe that our analysis is not circular.

To avoid confusions, we have also rephrased this section: "Levels of expression of SBG were comparable between sexual lineages within each species so that the observed pattern of defeminisation and masculinisation is specific to the transition to asexuality in Amazons of both species"

Reviewer #3 (Remarks to the Author):

This paper looks at phenotypic and genetic changes following the loss of sex in a brown algae. The paper addresses a key question in evolutionary biology, and is well written and analysed. I have only a few comments.

Firstly, it would be good to provide more information on the pheromone production. At line 441 you say it is complex, but up until then I thought it could be relatively simple? i.e. caused by a null mutation or something?

Re: We thank the reviewer for the positive comments. Concerning the complexity of the pheromone production pathway, we have added a sentence in the introduction about this (to prepare the reader) You could look in your data for this, either by looking for genes with null mutations in the amazons, or genes with zero expression in amazons but with high expression in females (I am not sure if you filtered these out?)

Re: We thank the reviewer for the suggestion, and indeed our comparative genomic analysis using male, female and amazon datasets tried to address this (we confirm we filtered the samples to find alleles that were associated with the amazon phenotype, i.e., genes that were expressed in all females but not in males or amazon individuals).

As we explain in the manuscript, we report at least one gene that is indeed associated with the capacity to produce the pheromone. To address the comment of the reviewer we have added more details in the Introduction about what is known about the pheromone, together with citations.

The authors have produced a lot of genetic resources (Genome assemblies, annotation, RNAseq), but their data does not seem to be available? In addition, they have performed complex analyses, but not provided their code. This should be provided either as supplementary material or in a digital repository (e.g. Zotero).

Re: The RNA-seq data are submitted to the NCBI SRA BioProject PRJNA999486; Details shown in table S1; NCBI SRA reviewer link:

<https://dataview.ncbi.nlm.nih.gov/object/PRJNA999486?reviewer=5ajgmsbf799vp17om0juunkooh>

The genomes including gene annotation will be available at <https://doi.org/10.17617/3.DBZ2C0> and <https://doi.org/10.17617/3.B5UHU0>

All data will be publicly available once the paper is accepted for publication.

Minor

Line 57 – I think that you should refer to Parker, D. J., Bast, J., Jalvingh, K., Dumas, Z., Robinson-Rechavi, M., Schwander, T. 2019. Repeated evolution of asexuality involves convergent gene expression changes. *Molecular Biology and Evolution*. 36: 350–364 rather than Parker, D. J., Bast, J., Jalvingh, K., Dumas, Z., Robinson-Rechavi, M., Schwander, T. 2019. Sex-biased gene expression is repeatedly masculinized in asexual females. *Nature Communications*. 10: 4638 here.

Re: we have replaced the reference as requested

Line 126 – ratio not ration

Re: This has been corrected

Fig 1C – please make it clear which lineages are being compared here.

Re: This has been done in the legend of the figure.

Line 195 – fully asexual – do you mean obligately asexual?

Re: yes, we changed the wording as suggested

Line 696 – gens should be genes.

Re: The typo has been corrected

Version 2:

Decision Letter:

23rd April 2024

Dear Susana,

Thank you for submitting your revised manuscript "PARALLEL LOSS OF SEX IN FIELD POPULATIONS OF A BROWN ALGA SHEDS LIGHT ON THE MECHANISMS UNDERLYING THE EMERGENCE OF ASEXUALITY" (NATECOLEVOL-24010297B). It has now been seen again by the original reviewers and their comments are below. The reviewers find that the paper has improved in revision, and therefore we'll be happy in principle to publish it in *Nature Ecology & Evolution*, pending minor revisions to satisfy the reviewers' final requests and to comply with our editorial and formatting guidelines.

Thank you again for your interest in *Nature Ecology & Evolution*. Please do not hesitate to contact me if you have any questions.

[REDACTED]

Reviewer #1 (Remarks to the Author):

The authors have thoroughly revised the manuscript, addressing the issues raised by all reviewers. The changes have resulted in a yet improved manuscript, which normally would be ready for publication. However, upon re-reading, I noticed an additional point, a potential statistical issue, which I think should be checked before publication (I apologize for not having spotted this earlier). It concerns the results on the de-feminization / masculinization of gene expression in Amazons. As discussed in the previous review and in the author's response, this results "does not add too much to the whole story". So even if it turns out to be an issue, the overall merits of the study should not be impacted. The point that should be checked is whether these results may be explained by a simple statistical effect, a "regression toward the mean". Indeed, the categories "female-biased genes" and "male-biased genes" may contain false positives, which then, when re-evaluated in Amazons, are logically expected to show more average expression levels (i.e., intermediate between males and females). Even among true positives, there is a risk that genes with exaggerated expression levels (due to random noise) have an increased likelihood of being identified as true positives (compared to other true positives). These genes can also be expected to show more average expression levels when re-evaluated in Amazons. The same potential issue occurs in male-biased genes. The issue arises mainly from the fact that

categories of genes are selected according to their extreme values in sexuals, and only these genes are then re-evaluated in Amazons. I am not entirely sure as to how to address this issue other than acknowledging it. However, an analysis that investigates the expression of all genes (i.e., without defining categories based on extreme values) may still address the question whether gene expression profiles of Amazons are intermediate between males and females or not.

A few minor issues remain also from the previous round of review, but I would leave it to the discretion of the authors to address them while preparing the final version of the manuscript:

L. 22-24 and L. 131: I still do not agree with this interpretation of the results (i.e., that they indicate decay of female functions rather than relaxation or release from sexual antagonism). The results do refute the idea that gene expression differences between males and females are less pronounced than "optimal" in sexual populations and that in the absence of males they are "allowed" to evolve more in the female direction. However, it is unclear why the presence of sexual antagonism should reduce male-female gene expression difference in the first place. Again, I know that this idea has been repeatedly put forward in the literature, but there are reasons to think that sexual antagonism may sometimes increase rather than decrease expression differences between sexes, e.g., due to genes specifically expressed in one sex to combat antagonism from the other sex. In short, I do not see a reason to repeat this argument (at least not without justifying it in more detail). I would recommend interpreting these results as evidence that gene expression is more strongly affected by the decay of traits mediating male-female interactions than by exaggeration of female traits in Amazon.

The new paragraph comparing results with animals (potential re-use and optimization of pre-existing asexuality pathways) lacks references. Parallels could be drawn here to findings for example in aphids and daphnia.

In my previous comment on investigating substitution rates in sporophyte-specific genes (comment on previous L. 377), I did not suggest comparing sporophyte-specific genes with gametophyte-specific genes. Rather, I suggested comparing sporophyte-specific genes between Amazons, where they are always 1n, and sexuals, where they are always 2n. This might be an interesting addition, but was not meant to replace the current analysis.

Reviewer #2 (Remarks to the Author):

The authors have addressed all my comments satisfactorily and I fully recommend publishing the manuscripts as it is. Congratulations on the very nice and interesting paper!

Reviewer #3 (Remarks to the Author):

The authors have addressed my queries well in their responses. This is a great paper.

Author Rebuttal letter:

Reviewer #1:

Remarks to the Author:

The authors have thoroughly revised the manuscript, addressing the issues raised by all reviewers. The changes have resulted in a yet improved manuscript, which normally would be ready for publication. However, upon re-reading, I noticed an additional point, a potential statistical issue, which I think should be checked before publication (I apologize for not having spotted this earlier). It concerns the results on the de-feminization / masculinization of gene expression in Amazons. As discussed in the previous review and in the author's response, this result does not add too much to the whole story. So even if it turns out to be an issue, the overall merits of the study should not be impacted. The point that should be checked is whether these results may be explained by a simple statistical effect, a regression toward the mean. Indeed, the categories "female-biased genes" and "male-biased genes" may contain false positives, which then, when re-evaluated in Amazons, are logically expected to show more average expression levels (i.e., intermediate between males and females). Even among true positives, there is a risk that genes with exaggerated expression levels (due to random noise) have an increased likelihood of being identified as true positives (compared to other true positives). These genes can also be expected to show more average expression levels when re-evaluated in Amazons. The same potential issue occurs in male-biased genes. The issue arises mainly from the fact that categories of genes are selected according to their extreme values in sexuals, and only these genes are then re-evaluated in Amazons. I am not entirely sure as to how to address this issue other than acknowledging it. However, an analysis that investigates the expression of all genes (i.e., without defining categories based on extreme values) may still address the question whether gene expression profiles of Amazons are intermediate between males and females or not.

Re: We acknowledged the potential issue in the methods section as suggested by the reviewer. We added the following sentence: "Although unlikely, we acknowledge that we cannot fully exclude a regression to the mean effect when analysing sex-biased genes in Amazons."

A few minor issues remain also from the previous round of review, but I would leave it to the discretion of

the authors to address them while preparing the final version of the manuscript:

L. 22-24 and L. 131: I still do not agree with this interpretation of the results (i.e., that they indicate decay of female functions rather than relaxation or release from sexual antagonism). The results do refute the idea that gene expression differences between males and females are less pronounced than optimal in sexual populations and that in the absence of males they are allowed to evolve more in the female direction. However, it is unclear why the presence of sexual antagonism should reduce male-female gene expression difference in the first place. Again, I know that this idea has been repeatedly put forward in the literature, but there are reasons to think that sexual antagonism may sometimes increase rather than decrease expression differences between sexes, e.g., due to genes specifically expressed in one sex to combat antagonism from the other sex. In short, I do not see a reason to repeat this argument (at least not without justifying it in more detail). I would recommend interpreting these results as evidence that gene expression is more strongly affected by the decay of traits mediating male-female interactions than by exaggeration of female traits in Amazon.

Re: we have rephrased the sentences to make them less strong, and put more accent on the decay of female functions

The new paragraph comparing results with animals (potential re-use and optimization of pre-existing asexuality pathways) lacks references. Parallels could be drawn here to findings for example in aphids and daphnia.

Re: We added a reference (Neimen et al, a review) as suggested.

In my previous comment on investigating substitution rates in sporophyte-specific genes (comment on previous L. 377), I did not suggest comparing sporophyte-specific genes with gametophyte-specific genes. Rather, I suggested comparing sporophyte-specific genes between Amazons, where they are always 1n, and sexuals, where they are always 2n. This might be an interesting addition, but was not meant to replace the current analysis.

Re: We thank the reviewer for the suggestion. However, we do not have data for Amazon sporophytes, so this analysis is not possible to do. As the reviewer agrees, this is a unnecessary addition, outside the scope of this manuscript.

Reviewer #2:

Remarks to the Author:

The authors have addressed all my comments satisfactorily and I fully recommend publishing the manuscripts as it is. Congratulations on the very nice and interesting paper!

Reviewer #3:

Remarks to the Author:

The authors have addressed my queries well in their responses. This is a great paper.

Version 3:

Decision Letter:

18th June 2024

Dear Susana,

We are pleased to inform you that your Article entitled "PARALLEL LOSS OF SEXUAL REPRODUCTION IN FIELD POPULATIONS OF A BROWN ALGA SHEDS LIGHT ON THE MECHANISMS UNDERLYING THE EMERGENCE OF ASEXUALITY", has now been accepted for publication in Nature Ecology & Evolution.

Over the next few weeks, your paper will be copyedited to ensure that it conforms to Nature Ecology and Evolution style. Once your paper is typeset, you will receive an email with a link to choose the appropriate publishing options for your paper and our Author Services team will be in touch regarding any additional information that may be required

Due to the importance of these deadlines, we ask you please us know now whether you will be difficult to contact over the next month. If this is the case, we ask you provide us with the contact information (email, phone and fax) of someone who will be able to check the proofs on your behalf, and who will be available to address any last-minute problems. Once your paper has been scheduled for online publication, the Nature press office will be in touch to confirm the details.

Acceptance of your manuscript is conditional on all authors' agreement with our publication policies (see

www.nature.com/authors/policies/index.html). In particular your manuscript must not be published elsewhere and there must be no announcement of the work to any media outlet until the publication date (the day on which it is uploaded onto our web site).

Please note that *Nature Ecology & Evolution* is a Transformative Journal (TJ). Authors may publish their research with us through the traditional subscription access route or make their paper immediately open access through payment of an article-processing charge (APC). Authors will not be required to make a final decision about access to their article until it has been accepted. [Find out more about Transformative Journals](https://www.springernature.com/gp/open-research/transformative-journals)

Authors may need to take specific actions to achieve [compliance](https://www.springernature.com/gp/open-research/funding/policy-compliance-faqs) with funder and institutional open access mandates. If your research is supported by a funder that requires immediate open access (e.g. according to [Plan S principles](https://www.springernature.com/gp/open-research/plan-s-compliance)) then you should select the gold OA route, and we will direct you to the compliant route where possible. For authors selecting the subscription publication route, the journal's standard licensing terms will need to be accepted, including [self-archiving and license to publish](https://www.nature.com/nature-portfolio/editorial-policies/self-archiving-and-license-to-publish). Those licensing terms will supersede any other terms that the author or any third party may assert apply to any version of the manuscript.

We welcome the submission of potential cover material (including a short caption of around 40 words) related to your manuscript; suggestions should be sent to Nature Ecology & Evolution as electronic files (the image should be 300 dpi at 210 x 297 mm in either TIFF or JPEG format). Please note that such pictures should be selected more for their aesthetic appeal than for their scientific content, and that colour images work better than black and white or grayscale images. Please do not try to design a cover with the Nature Ecology & Evolution logo etc., and please do not submit composites of images related to your work. I am sure you will understand that we cannot make any promise as to whether any of your suggestions might be selected for the cover of the journal.

You can generate the link yourself when you receive your article DOI by entering it here: <http://authors.springernature.com/share>.

[REDACTED]

P.S. Click on the following link if you would like to recommend Nature Ecology & Evolution to your librarian <http://www.nature.com/subscriptions/recommend.html#forms>

** Visit the Springer Nature Editorial and Publishing website at http://editorial-jobs.springernature.com?utm_source=ejp_NEcoE_email&utm_medium=ejp_NEcoE_email&utm_campaign=ejp_NEcoE for more information about our career opportunities. If you have any questions please click [here](mailto:editorial.publishing.jobs@springernature.com).
